



# Investigating hydroclimatic impacts of the 168-158 BCE volcanic quartet and their relevance to the Nile River basin and Egyptian history

Ram Singh[1,2], Kostas Tsigaridis[1,2], Allegra N. LeGrande[2,1], Francis Ludlow[3], Joseph G Manning[4]

[1] Center for Climate System Research, Columbia University, New York
[2] NASA Goddard Institute for Space Studies, New York, NY-10025
[3] Department of History, School of Histories and Humanities, Trinity College, Dublin 2, Ireland
[4] Departments of History and Classics, Yale University, New Haven, CT 06520, USA

*Correspondence to*: Ram Singh (rs4068@columbia.edu)

**Abstract.**

The Ptolemaic era (305-30BCE) represents an important period of Ancient Egyptian history known for its major material and scientific advances, but also ongoing episodes of political and social unrest in the form of (sometimes widespread) revolts against the Ptolemaic elites. While the role of environmental pressures has long been overlooked in this period of Egyptian history, ice-core-based volcanic histories have identified the period as experiencing multiple notable eruptions, and a repeated temporal association between explosive volcanism and revolt has recently been noted. Here we analyze the global and regional (Nile River Basin) climate response to a unique historical case of 4 consecutive and closely timed eruptions (first a tropical one, closely followed by 3 extratropical northern hemispheric events) between 168 and 158 BCE, a particularly troubled period in Ptolemaic history for which we now provide a more detailed hydroclimatic context. The NASA GISS ModelE Earth system model simulates a strong radiative response with a radiative forcing (Top of atmosphere) of -7.5 W/m2 (following the first eruption) and -4.0 w/m2 (after each of the 3 remaining eruptions) at a global scale. Associated with this, we observe a global cooling of the order of 1.5°C at the surface following the first (tropical) eruption, with the following three extratropical eruptions extending the cooling period for more than 15 years. Consequently, this series of eruptions constrained the northward migration of the inter-tropical convergence zone (ITCZ) during the northern hemisphere summer monsoon season, and major monsoon zones (African, South and



East Asian) experienced suppression of rainfall >1 mm/day during the monsoon (JJAS) season averaged for 2 years after each eruption. A substantial suppression of north African and Indian summer monsoon over the Nile River headwater region vigorously affects the river flow in the catchment and river discharge. River mass flow consecutively decreases by up to more than 30% relative to an unperturbed from volcanoes annual mean flow for 2 years after the tropical eruption. A moderate decrease of up to 15-20% is produced after each of the remaining eruptions. These results show that the first eruption produces a strong hydroclimate response, and the following 3 eruptions prolonged the drying conditions. These results also support the contention that the observed association between ice-core-based signals of explosive volcanism and the hydroclimatic impact of these eruptions during the Ptolemy era, including the suppression of the critical for agriculture Nile summer flooding.

**Key Words:** Volcanic eruption, hydroclimate impacts, Inter-tropical Convergence Zone (ITCZ), Monsoon, Nile River basin

## 1. Introduction

Stratovolcanic eruptions that result in high altitude sulfate aerosol distribution across one or both hemispheres can diminish insolation, leading to both global and regional impacts on climate and people (Robock 2000; Toohey et al 2019). The cooling caused by such events reduces evaporation (and thus precipitation; Lui et al. 2016; Iles et al 2013), while also potentially leading to a near global-scale dynamical suppression of the boreal summer northward migration of the inter-tropical convergence zone (ITCZ), as the convergence follows the surface area of maximum temperature (Petterson et al. 2000; Chiang and Bitz 2005; Broccoli et al., 2006). Furthermore, these changes in precipitation can impact river outflow (Oman et al. 2006; Sabzevari et al., 2015; Kostiç et al 2016), which has implications for civilizations from antiquity to the present-day, not least by affecting food security (Manning et al. 2017). The Nile River, upon which Egyptian civilization was heavily dependent, is no exception. With ice-core-



based volcanic forcing histories now identifying hundreds of potentially climatically effective eruptions over the past several millennia (Sigl et al. 2015), Egyptian civilization may have been repeatedly influenced by the "hydroclimatic shocks" wrought by these events (Manning et al. 2017).

Explosive volcanic eruptions are the major natural source of forced variability in the climate system at yearly to decadal time scales (Schmidt et al., 2011; Colose et al., 2016; Swingeduow et al., 2017; Khodri et al. 2017). Powerful stratovolcanoes can inject sulfur-rich gases high into the stratosphere, where they oxidize to form sulfate aerosol particles that can persist for months to years, in turn impacting the climate on regional to global scales. Volcanic aerosols in the stratosphere cause cooling in the troposphere by scattering incoming shortwave radiation, while also heating the stratosphere (Robock and Mao, 1992). Unequal north-south stratospheric heating due to volcanic aerosol presence concentrated in lower latitudes after tropical eruptions can influence major modes of atmospheric circulation and surface climate variability such as the Arctic Oscillation/North Annular Mode (AO/NAM) and North Atlantic Oscillation (NAO), in effect by driving an enhanced westerly airflow (Shindell et al., 2004; Zanchettin et al., 2021). The post-volcanic surface temperature response can also affect the El Niño/La Niña and Southern Oscillation, as well as having a long-term impact on Atlantic Meridional Overturning Circulation (AMOC) strength (Khodri et al., 2017; Wahl et al., 2014; Robock and Mao, 1995; Pausata et al., 2015). Volcanic injection of sulfur-containing compounds can, too, influence stratospheric chemistry, yielding further complex atmospheric and climatic responses upon interacting with water and halogens (LeGrande et al., 2016; Brenna et al., 2020; Staunton-Sykes et al., 2021)). Paleo records and climate modeling efforts suggest that the dynamical response of volcanic aerosol causes a net (but regionally variable) drying effect and significantly impacts the global rainfall pattern (PagesHydro2k/Smerdon et al., 2017; Colose et al., 2016; Liu et al., 2016; Iles and Hegerl, 2014). For example, Trenberth and Dai (2007) analyzed the impact of the Pinatubo (1991) eruption on land precipitation and river streamflow and found an increase in associated drought conditions after the eruption in 1992. Joseph and Zeng (2011) suggested that varying responses to volcanically induced rainfall anomalies over land and ocean can seasonally modulate drought conditions in the tropics. In addition, hemispherical asymmetrical radiative forcing due to biases in the distribution of volcanic aerosols creates a radiative imbalance across the hemisphere, impacts the



movement of ITCZ, constraining the extent of its summertime migration into the energetically deficit hemisphere (Colose et al., 2016; Xian and Miller, 2008).

Volcanic eruptions resulting in an asymmetrical latitudinal aerosol burden (e.g., the Katmai eruption in 1912 and El Chichón in 1982) are thought to have enhanced the 20th century Sahelian drought by shifting the surface temperature maxima and influencing the strength and position of Hadley cells (Haywood et al 2013). The Laki fissure eruption series (1783/84) injected approximately 122 Mt of $SO_2$ into the atmosphere over eight months and produced a strong cooling and suppression of African monsoon (Oman

et al., 2006; D'Arrigo et al., 2011), resulting in reduced Nile River flooding, or what is known colloquially as "Nile failure". Similar impacts were simulated over the African region for the Katmai (1912) eruption (Vorosmarty et al., 1998; Thordarson and Self, 2003; Oman et al., 2005; Oman et al., 2006). African monsoon rainfall over the Ethiopian highlands contributes (mainly via the Blue Nile and Atbara River) ~85% to the Nile summer flood over the Egyptian plains and is a strong control over the associated

interannual variability of the flood (Melesse et al., 2011). Before the construction of large dams in the twentieth century, a failure of the African monsoon was thus historically associated with insufficient water to extensively practice the flood recession agriculture that ordinarily delivered such high agriculturally productivity in the Nile valley, and for which ancient Egypt was famed, often leading to adverse societal impacts (e.g., Butzer, 1976; Hassan, 1997; Hassan et al., 2007).

One of the most richly documented periods of ancient Egyptian history is the Ptolemaic era, 305-30 BCE, during which time Egypt was ruled by Greeks in a lineage beginning with Ptolemy I Soter (d. 283 BCE), who had been one of Alexander the Great's key generals and instrumental in the conquest of Egypt. The period distinguishes itself through its mixing of Greek and Egyptian traditions and its great material, cultural and scientific achievement (not least in the founding of the city of Alexandria on the

Mediterranean coast with its famed Great Library and Lighthouse), but also through its chronic political instability through time (Ludlow and Manning, 2016; 2021). Historical records attest to dynastic power struggles and repeated revolts against Ptolemaic rule that have, until recently, been mainly credited to the poor quality of Ptolemaic leadership, particularly following the death of Ptolemy III in 222 BCE (McGing, 1997; Veïsse, 2004). Little consideration has been given to external environmental influences, despite the

great dependence of Egyptian agriculture on the Nile summer flood. Recent chronological corrections to



ice-core-based volcanic forcing histories for the Ptolemaic period (Sigl et al., 2015) have, however, revealed a close correspondence between the timing and frequency of revolts and inferred-tropical and NH extratropical explosive eruptions (Ludlow and Manning, 2016, 2021; Manning et al., 2017), implying a previously unrecognized role for volcanism in the turbulent history of the kingdom. An example is the

"Great Theban Revolt" of c.207 BCE, occurring shortly after a notable 209 BCE tropical eruption, when the Ptolemies lost control of large areas of the Nile Valley to a sequence of two apparently native Egyptian Pharaohs (Sigl et al., 2015; Ludlow et al., 2022).

Here, the fundamental linkage involves the societal response to sudden hydroclimatic shocks wrought by the widespread northern hemispheric cooling that can follow major explosive volcanic eruptions and

which can act (as noted above) to reduce the meridional (north-south) temperature contrast that drives the African monsoon. When this leads to a "failure" of the agriculturally critical Nile summer flood, a range of societal impacts can be expected, most obviously agricultural and economic, with reduced food security for families who may also have been less able to meet state taxation demands, potentially necessitating the sale of their hereditary lands and prompting migration from rural areas to larger urban areas in search

of food (Manning, 2003; Manning et al., 2017). This would likely compound the psychological, religious and, ultimately, political significance of a "failed" Nile flood, with such an event being widely feared among the general populace and with the potential to be interpreted (and propagandized) as a reflection of divine displeasure at the Pharoah (Ludlow and Manning, 2021; Ludlow et al., 2022). In the context of a period when parts of the populace, including at least some of the older native Egyptian elites and

priesthood, were likely resentful of Greek rule and the taxation and other advantages given to those of Greek backgrounds (McGing, 1997; Ludlow et al., 2022), a Nile failure may have held additional political potency, helping to explain the repeated link between ice-core-based eruption dates and major revolt onset.

Important questions remain, however, in particular on the role of hydroclimatic shocks in the longer-term

declining stability of the state and its ability to project power across the eastern Mediterranean. Repeated revolt in the third to first centuries BCE speaks to persistent vulnerability, yet despite experiencing the hydroclimatic effects of multiple eruptions (including those with a greater climate forcing potential than has been experienced in the twentieth and twenty-first centuries (Sigl et al., 2015)) and multiple such



revolts, the dynasty persisted for almost three centuries, simultaneously suggesting a considerable level
of resilience. It is conspicuous, however, that the dynasty ultimately ended (with Cleopatra's defeat by
Rome at the naval battle of Actium in 31 BCE and her suicide in 30 BCE) in a decade that followed one
of the largest explosive eruptions of the last 2,500 years in terms of climate forcing potential (based upon
polar ice-core sulfate deposition levels), that of Okmok (Alaska) in early 43 BCE (McConnell et al.,
2020). This itself followed a smaller but notable (likely extratropical NH eruption) in 46 BCE (McConnell
et al., 2020). Egypt in the 40s BCE had, perhaps unsurprisingly therefore, experienced repeated Nile
failure, famine, plague, inflation, administrative corruption, rural depopulation, migration, and land
abandonment (Hölbl, 2001; Roller, 2010). It is notable though that there is no convincingly documented
revolt, perhaps owing to Cleopatra's abilities as a leader and interventions in grain distribution to prevent
starvation of the population. The stresses of the 40s BCE may still, however, be credibly posited as
weakening Egypt's hand against Rome as it became entangled in the complex political and military
developments of this major moment in world history, as Rome transitioned from its republic to imperial
form (Manning et al., 2017; McConnell et al., 2020; Ludlow and Manning, 2021).

Here clearly, as at any other time, the societal impact of a given hydroclimatic shock will be mediated by
the prevailing historical context, but this does not mean all hydroclimatic shocks are of equal potential
impact. Thus, regarding the association between explosive volcanism and Chinese dynastic collapse over
the Common Era, the societal efficacy of volcanic climate forcing was observed to depend not only on
levels of pre-existing or contemporaneous societal stress or instability (i.e., the historical context), but
also on the magnitude of the climate forcing itself (Gao et al., 2021). Ice-core data suggest that the
sequence of approximately 24 tropical and extratropical NH eruptions experienced by Ptolemaic Egypt
varied in their climate forcing potential and were not distributed evenly in time (Sigl et al., 2015; Manning
et al., 2017). Clusters of historical eruptions have at other times been examined for their potentially severe
climatic and societal impacts (e.g., Toohey et al., 2016; Guillet et al., 2020; Campbell and Ludlow, 2020;
Stoffel et al., 2022) and such an investigation can make a meaningful contribution to our understanding
of the role of explosive volcanism in the history of Ptolemaic Egypt. A time of particular intertest is the
160s BCE, a decade of considerable internal revolt and instability. Indeed the Ptolemaic dynasty might
well have fallen here if not for self-interested Roman intervention against the Seleukid empire (great



rivals to the Ptolemies), after their successful invasion (170-168 BCE) of Egypt under the command of Antiochus IV (Grainger, 2010; Blouin, 2014; Manning et al., 2017). This is also a decade remarkable for three notable volcanic eruptions (168, 164 and 161 BCE), with a further event in 158 BCE (Sigl et al., 2015).

While high-resolution palaeoenvironmental proxies for Egypt are effectively absent in this early period, our understanding of the hydroclimatic impacts of the sequence of eruptions between 168 and 158 BCE can be advanced by climate modeling. The distribution of sulfate across both poles (Sigl et al., 2015) identifies the first eruption (168 BCE) as the largest and likely occurring in the tropics, followed by three equally separated and comparably moderate-sized extratropical eruptions in the northern hemisphere. No previous study has specifically explored such a set of four closely consecutive eruptions or their impacts on the regional hydroclimate of a major ancient-era civilization. The few previous studies that have thus far examined the climatic and societal effects of eruption clusters include an exploration of the volcanic event cluster of the early 12th century (between 1108 and 1110 CE) (Guillet et al., 2020), the double event of the 6th century in 536 and 540 CE (Toohey et al., 2016), perhaps better seen as a triple event, in view of the additional, if much smaller, eruption in 546 (Sigl et al., 2015), and the eruption cluster from 1637 to 1641 (Stoffel et al., 2022). These studies have variously employed palaeoclimatic data, written evidence and/or climate model simulation to reveal the strong negative temperature anomalies over the Northern hemisphere following these eruptions, thereby suggesting the potential for adverse effects on crop yields and providing a climatic context by which to better understand the human history of these periods (Guillet et al. 2020; Toohey et al. 2016; Stoffel et al., 2022).

In this study, we thus use a computationally expensive but more sophisticated version of the National Aeronautics and Space Administration (NASA), Goddard Institute for Space Studies (GISS) Earth system model, GISS ModelE, to simulate the series of eruptions from 168 to 158 BCE and analyze the impacts on regional hydroclimate over the Nile River basin. GISS ModelE2.1-MATRIX is the version of ModelE that has interactive chemistry and aerosol microphysics (MATRIX; Bauer et al., 2008, Bauer et al., 2020). Details of the model and methodology employed to conduct the experiment and analysis are discussed in section 2. Our estimation of the background climate of the 2.5k (orbital and greenhouse gases (GHGs) changes only), together with the impacts due to PMIP4 vegetations are analysed and discussed in under





section 3. Further, sub-sections under the section 3 also focus on the evaluating the NASA GISS ModelE simulated volcanic aerosol properties during this period and analyzed the radiative impacts of volcanic aerosols due to this set of 4 eruptions. These results focus on analyzing the modelling capability to resolve the microphysical properties of volcanic aerosols, aerosol optical depth and evaluation of aerosol size, which controls the radiative and the climatic impacts of volcanic eruptions (Timmreck et al 2009; Schmidt

et al. 2010). This is important because enormous volcanic eruptions may not have comparatively more significant climate impacts due to higher collision, larger size and affected radiative feedback (Timmreck et al., 2010). In section3, we also focus on analyzing the hydroclimatic impacts of volcanic quartet over the Nile basin region. Finally, the discussion and conclusion section summarize the results and consider how they can advance our understanding of the period's fraught human history in Egypt. This case of 4

consecutive eruptions presents a unique case to study the role of multiple eruptions on regional climate over such a critical region (Nile Basin).

## 2. Methodology & Experiment design

### 2.1 Model Description

We used the NINT (Non-INTeractive) version (Kelley et al., 2020) of GISS ModelE2.1 to simulate the background climate conditions corresponding to the period 2.5ka years BP. The term non-interactive means that atmospheric composition and climate are decoupled, so any changes in composition are handled by external input only. After reaching equilibrium, we enabled atmospheric composition-climate interactions for the experiments performed here, as described below.

GISS ModelE2.1 is a state-of-the-art Earth System Model contributing to the Climate Model Intercomparison Project (CMIP) phase 6 (Eyring et al., 2016). The atmospheric component of GISS ModelE2.1 simulates on a horizontal resolution of 2° latitude by 2.5° longitude with 40 vertical layers and a model top at 0.1 hPa. It is coupled to the GISS Ocean v1 model at horizontal resolution of 1° latitude by 1.25° longitude with 40 layers. The Demographic Global Vegetation Model (DGVM) is Ent Terrestrial

Biosphere Model (TBM) (Kiang, 2012; Kim et al., 2015) is used to implement the climate-controlling vegetation properties, including the satellite-driven (MODIS) plant functional types (PFTs) and monthly



varying leaf area index (LAI) (Gao et al., 2008; Myneni et al., 2002). The tree heights come from Simard et al. (2011) and include the carbon cycle interactively (Ito et al., 2020). The MATRIX (Multiconfiguration Aerosol TRacker of mIXing state) aerosol microphysics module (Bauer et al., 2008; Bauer et al., 2020) is used in the coupled composition-climate runs described here to simulate the active volcanism and corresponding climate conditions. MATRIX is an aerosol microphysics scheme using the quadrature method of moments, representing new particle formation (Vehkamaki et al., 2002), aerosol-phase chemistry, condensational growth, coagulation and mixing state of aerosols (Bauer et al., 2013). MATRIX tracks 16 mixing states, 51 aerosol tracers and resolves mixtures of sulfate, nitrate, ammonium, aerosol water, black carbon, organic carbon, sea salt and mineral dust (Bauer et al., 2008). MATRIX includes the direct effect and the first indirect effect of aerosols on climate.

## 2.2 Experiment Design

A control simulation for the 2.5ka period is performed using the PMIP4 (Paleoclimate Model Intercomparison Project) phase 4 protocols for the mid-Holocene (6ka) experiment, altered for conditions appropriate to 2.5ka. These include altering the orbital forcing, greenhouse gases ($CO_2$: 279 ppm, $N_2O$: 266 ppb, and $CH_4$: 610 ppb), as well as the vegetation in Africa and the high boreal Eurasia and North America (Otto-Bliesner et al., 2017). Ozone and aerosols are prescribed to non-anthropogenic conditions only. The orbital and greenhouse gas forcings for the 2.5ka period are expected to play a vital role in producing the correct equilibrium climate. We ran a control run with the NINT configuration for 1000 years to get the model in equilibrium, and then extended for 100 years by adding the MATRIX version of ModelE2.1 to again achieve an equilibrium state for a 2.5ka period with composition-climate interactions turned on. Vegetation cover, LAI and vegetation height are prescribed corresponding to the piControl period climate. The unavailability of exact vegetation cover information for the relevant period restrains the GCMs without dynamic vegetation model to produce mid-Holocene warm Northern hemisphere summer and enhanced NH monsoons conditions (Tierney et al, 2017; Larrasoaña et al, 2013). However, the vegetation cover used here as defined by the PMIP4 protocol (Otto-Bliesner et al., 2017) for the mid-Holocene period shows an intense impact on North-African rainfall and explains the difference between simulated and reconstructed climate conditions (Braconnot et al., 1999; Pausata et al.,





2016). For this, we created a modified mid-Holocene boundary condition sensitivity vegetation map using
the following postulates and linearly interpolating between 6ka and the preindustrial (PI) for the 2.5ka
period.

- Northern hemisphere high latitude tundra during the preindustrial is replaced by boreal forests in 6ka.
- Sahara in the preindustrial is replaced by evergreen shrubs up to 25N and further north with savanna/steppe in 6ka.

Fig S1 (Supplementary information) shows the major vegetation plant function type (PFTs) cover changes under the PMIP4 sensitivity vegetation protocols after linearly interpolating for 2.5ka period.

The 2.5k equilibrated simulation with MATRIX is then extended for 70 more years with a corrected dust tuning, a typical process when equilibrating the model on a new climate state, and further 130 years with the linearly interpolated PMIP4 vegetation described above (refer to table TS1 for details of control runs and annual global mean time series of surface air temperature and precipitation in Fig. S2). This run
equilibrated very quickly and no further tuning was needed, so we used the last 100 of the total 130 years of that equilibrated run as the base climate to our analysis. An ensemble of 10 members with active volcanic eruptions was simulated using a restart file every 10 years during the last 100 years of the control simulation corresponding to 2.5ka period as summarized in table TS1, following the same approach as performed for the CMIP6 ensemble simulations (Kelley et al., 2020). The starting timepoint for each
ensemble member is shown by blue vertical lines in fig S2. Each ensemble member started on January 1[st] of the year 169 BCE and ran for 16 years, with each eruption happening on the 15[th] of June of the 2[nd], 6[th], 9[th] and 12[th] years.

An approximate eruption location is important in order to accurately estimate volcanic impacts on the climate system (Toohey et al., 2016, Aquila et al., 2018 [https://acd-
ext.gsfc.nasa.gov/Documents/NASA_reports/Docs/VolcanoWorkshopReport_v12.pdf]). The broad hemispheric position of the 168 to 158 BCE series of eruptions is thus chosen (to begin) with reference



to the bi-polar multi-ice-core sulphate deposition data of Sigl et al. (2015), which allows a discrimination between likely tropical (low-latitude) eruptions and those likely occurring in the extratropics of either hemisphere. Without any firm additional data (e.g., ice-core tephra) indicative of a more precise location, however, the ultimate location is by necessity selected more arbitrarily. The chosen locations of all eruptions are shown in Fig S3. We note, however, that the longitude of each eruption is not expected to play a major role as an uncertainty factor. The forcing potential of these four eruptions in terms of $SO_2$ injected into the atmosphere is also estimated using the Sigl et al. (2015) multi-ice-cire record of sulfate deposition over Greenland and Antarctica, linearly scaled corresponding to Pinatubo eruption estimates of 18.5 Tg $SO_2$ (Wolfe and Hoblitt, 1996). The injection height is selected to match that of Pinatubo, in the absence of any further information.

Table 1. Details of eruptions applied for this experiment with each eruption happening on the 15th of June of the 2nd, 6th, 9th and 12th years.

| Eruption | Year (BCE) | Position | Eruption injection ($SO_2$) | Injection Height (km) |
|---|---|---|---|---|
| E1 | 168 | Pinatubo (15.13, 120.35) (Tropical) | 22.5 Tg | 22-26 |
| E2 | 164 | Mt Laki, Iceland (64.03, -18.13 W) (NH) | 6.5 Tg | 22-26 |
| E3 | 161 | Mt Katmai, Alaska Peninsula (58.28, -154.95 W) (NH) | 7.2 Tg | 22-26 |
| E4 | 158 | Shiveluch, Kamchatka, Russia (56.39,161.21) (NH) | 7.5 Tg | 22-26 |





## 3. Results

### 3.1.1 2.5Ka GHG+ORB climate

We compared the 2.5ka equilibrium climate with only GHG and orbital forcing changes against a preindustrial (year 1850) control run to evaluate the impact of orbital and greenhouse gas changes alone on our base climate state. Surface air temperature shows no noticeable differences except a warming of northern hemisphere high latitudes due to the different orbital forcing (Fig 1). The northern hemisphere monsoon season (JJAS) rainfall slightly decreases along the northern equatorial belt. This points to the

limitation of the GISS model in not having an interactively dynamic vegetation component to reproduce the mid-Holocene wet African land cover (Harrison et al., 2015). Numerous studies have demonstrated that both biogeophysical feedback processes and atmospheric dynamics helps in achieving the wet African conditions for mid-Holocene (Kutzbach et al., 1996; Claussen et al., 2003; Kutzbach and Liu., 1997; Hewitt and Mitchell, 1998). Using the PMIP4 vegetation over the northern hemisphere regions has

been shown to provide a solution to a long-standing issue with CMIP3/CMIP5 models failing to reproduce wet African conditions for mid-Holocene (Harrison et al., 2015).

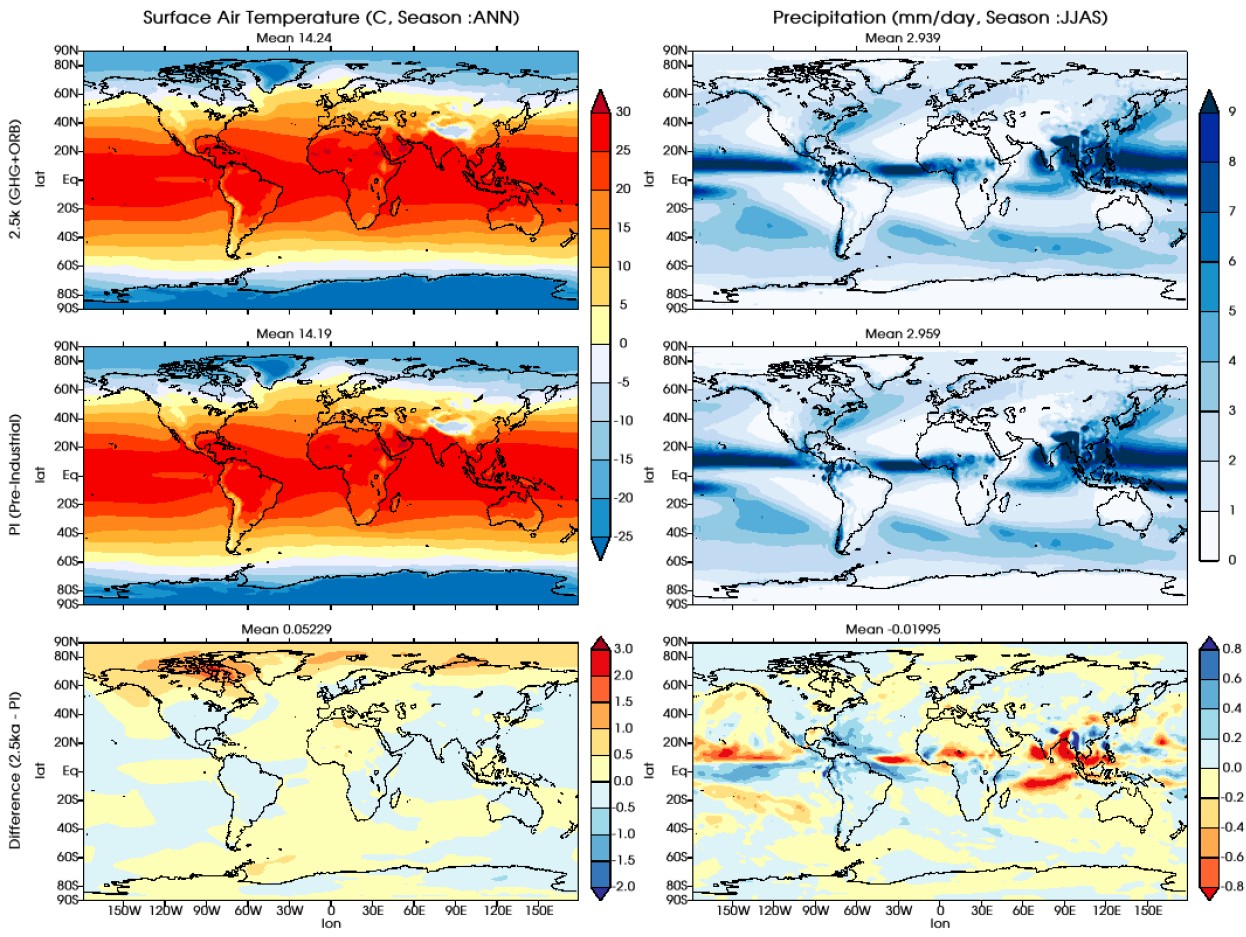

Fig 1. Annual mean of surface air temperature (left) and monsoon (JJAS) season mean precipitation
(right) for the equilibrium runs with orbital and GHG concentration changes only for the 2.5k period (top),
the preindustrial period (middle), and their difference (2.5ka-preindustrial; bottom) as simulated by GISS
ModelE2.1.

### 3.1.2 2.5Ka ORB+GHG+VEG climate

The comparison of mean climate for the 2.5ka period for inclusion of PMIP4 vegetation is shown in fig
2 for the annual mean surface air temperature and precipitation for the northern hemisphere monsoon
(JJAS) season.







Fig 2. Annual mean of surface air temperature (left) and monsoon (JJAS) season mean precipitation (right) for the equilibrium runs without (top row) and with (middle row) the PMIP4 vegetation for 2.5k period as simulated by GISS ModelE2.1. The difference of the two is shown at the bottom. We used a short initial notation for forcing to denote the difference (ORB+GHG+VEG = OGV and ORB+GHG= OG)


GISS ModelE2.1 simulates a global annual mean surface air temperature (SAT) of 14.4 ºC for the 2.5ka (ORB+GHG+VEG) simulation, which is 0.11 C higher than the 2.5ka (ORB+GHG) simulation without the vegetation changes. A strong increase in the surface air temperature of the order of 2-3 ºC is calculated over the northern hemisphere high latitude land regions, particularly in areas where land cover is replaced



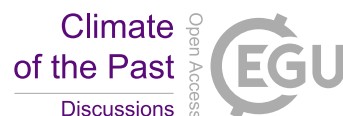

by boreal forest, decreasing ground albedo during snowy winter months, and a moderate rise of 0.5 ºC over Africa, which coincides with the regions of vegetation changes as described in section 2.2. The regional pattern of difference in the northern hemisphere monsoon season (JJAS) is mostly over the North African and Asian regions. The observed increase of 0.4 mm/day or greater over the North African and Southwest Asian monsoon regions suggests a northward movement of the ITCZ during the monsoon

season that is consistent with expectations given the modified vegetation for this period is consistent with our current understanding of mid-Holocene rainfall regimes (Tierney et al., 2017).

We also analyzed the zonal changes of longwave and shortwave radiation at the top of the atmosphere with ground albedo, as shown in Fig 3. The vegetation-albedo feedback due to the inclusion of woody forest on higher latitudes and shrubs and steppes over northern Africa plays a crucial role in the additional

monsoon season rainfall over the North African region. Greater vegetation cover for the Sahara and at higher latitudes in the Northern hemisphere alters the ground albedo by more than 10% regionally as well as altering the absorption of incoming solar radiations across the northern hemisphere latitudes (Fig 3). Consequently, it raises the pole-equator temperature gradient and pulls the ITCZ northwards as shown in Fig 2. We concluded that the control climate generated using the PMIP4 vegetation scaled from the mid-

Holocene to 2.5k period provides more precise and suitable control conditions to investigate the climatic impact of forcing perturbations due to volcanic eruptions. Vegetation boundary conditions according the PMIP4 sensitivity experiments with orbital and greenhouse gas forcing help in producing a precise equilibrium climate condition for this historically and climatically important period 2.5ka years ago.

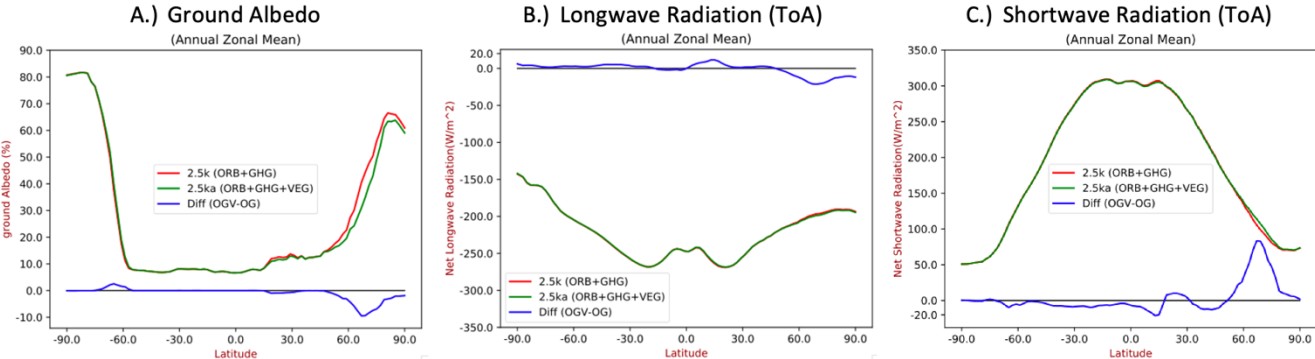

Fig 3. Annual zonal mean of ground albedo (A), longwave & shortwave radiation at the top of the atmosphere (ToA; B and C) for 2.5ka climates with ORB+GHG and ORB+GHG+VEG (red and green





line) respectively. The blue line shows the difference between them. The blue line in panel B & C is 10-fold (x10) to the original difference values in order to clearly show the difference on the same vertical axis.


## 3.2. Radiative forcing and climate response to volcanic aerosols

We simulate a series of four eruptions all occurring mid-June, during the 2nd, 6th, 9th and 12th years of the simulation, as described in section 2.2 and Table 1. Explosively injected $SO_2$ forms aerosols in the stratosphere that can then alter the radiative balance at the top of the atmosphere by scattering incoming

solar radiation and absorbing and re-emitting longwave radiation. Fig 3 shows the different components of the radiative budget on a monthly scale, with the annual cycle climatology removed for the entire period covering all four eruptions. The relative impacts of scattering the shortwave (SW) and absorbing the longwave (LW) radiation is proportional to the sulfate aerosol size (Lacis, 1992). The model simulated a lifetime for volcanically injected $SO_2$ as $31.4\pm0.72$ days for eruption E1 and $24.4\pm0.44$, $25.02\pm0.40$ and

$25.5\pm0.36$ days for eruption E2, E3 and E4, respectively. Other studies have reported a comparable average lifetime of 33 days (Read et al., 1993), $25\pm5$ days (Guo et al., 2004), 35 and 25 days (Bluth et al., 1992; Schnetzler et al., 1995) for $SO_2$ injected from 1991 Pinatubo eruption using various satellite retrievals.



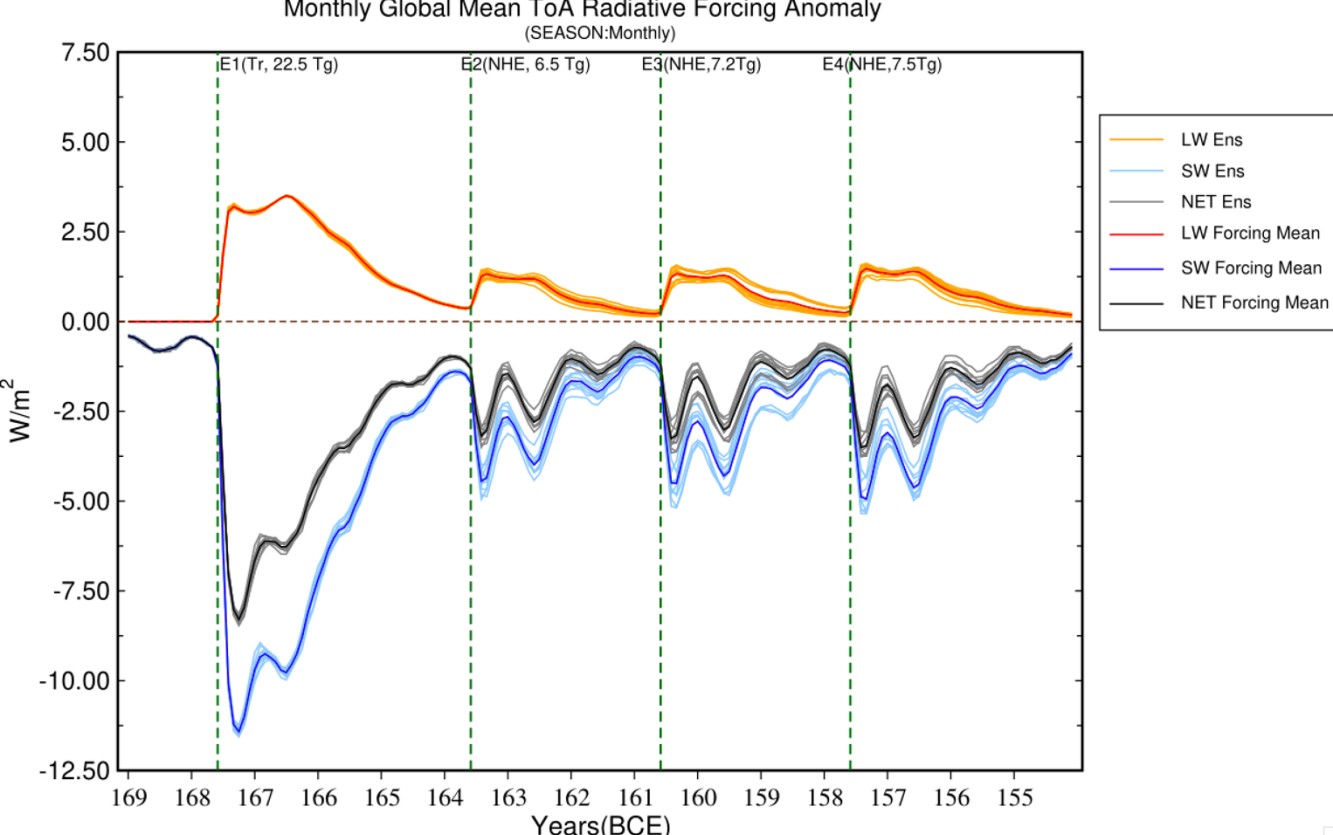


Fig 4: Monthly mean top of the atmosphere radiative balance perturbation due to volcanic aerosols for the entire simulation length. Orange/red shows the longwave radiative response, light/dark blue represents the shortwave, and grey/black represents the net (ToA) radiative change averaged at the global scale. The light-colored solid lines represent individual ensemble members, and the dark colored lines show the ensemble mean. The green vertical dashed lines show when the eruptions happened.

With 22.5 Tg of $SO_2$ injected, the first, tropical eruption was larger than Pinatubo (~30%) and altered the longwave radiation budget by a mean of ~3 $W/m^2$ for almost a year after the eruption, while the other three eruptions were approximately 1/3rd of Pinatubo and produced a perturbation of the longwave radiative budget by ~1 $W/m^2$. The model simulates a strong impact on the shortwave radiation budget up to a mean of ~10 $W/m^2$ for a few months after the first eruption and of ~4 $W/m^2$ for a few months after each of the subsequent eruptions. A mean imbalance of up to 8 $W/m^2$ after the first eruption and 2.5-3 $W/m^2$ after the other eruptions in the top of atmosphere net radiative forcing suggest strong cooling at





surface. The bumps in various radiative forcing trajectories in the year after the eruptions reflect the

seasonal cycle of sun in the northern hemisphere. The presence of volcanic aerosols in the atmosphere

altered climate in several ways, as described below.



Fig 5. Globally averaged changes in MSU TLS (top panel), surface air temperature (middle panel) and total column AOD at 550 nm for each month for the entire simulation period. The light-colored solid lines

represent individual ensemble members, and the solid dark colors show the ensemble means. The green vertical dashed lines show when the eruptions happened.

The top panel in Fig. 5 shows the monthly change in microwave sounding unit (MSU) temperature for the lower stratosphere (TLS) as calculated by the model, which is a typical metric for present-day





evaluation of modeled stratospheric temperatures against satellite data. It covers the lower stratosphere,
where volcanic aerosols mostly lie, and represents the local atmospheric response of longwave absorption
by them. After the Mount Pinatubo (1991) eruption, a lower stratospheric warming of the order of 2-3˚C
for a year is estimated using multiple reanalysis products (Labitzke and McCormick, 1992; Fujiwara et
al., 2015). This is comparable to the somewhat larger eruption simulated here, E1, in which volcanic

aerosols spread over a larger region in the northern and southern hemispheres and absorb a significant
portion of longwave radiation, warming the lower stratosphere by up to 3ºC for the first two years after
the eruption. This effect intensifies during the second year, before starting to steadily decline in Years 3
and 4 with the scavenging of volcanic aerosols. The other three eruptions warm the lower stratosphere by
up to 0.5º C only, because these were both weaker and extratropical eruptions that only affected the

northern hemisphere for a shorter period (~18 months). The lag in global mean surface air temperature
response (Middle panel) is mostly due to the thermal inertia of ocean which needs more time to return to
the normal as shown in the supplementary information (Fig S4).

The lower panel in Fig 5 presents the aerosol optical depth (AOD), a measure of atmospheric opacity to
the incoming radiation as the extinction (sum of scattering and absorption) of shortwave radiation at

550nm. The model simulates an AOD anomaly of around 0.21 for the first 18 months after the first
eruption, which decreases as aerosols are being removed. The subsequent eruptions produce an AOD of
the order of ~0.1 which similarly decreases with time. For comparison, the AOD estimation for the
Pinatubo (1991) eruption is 0.15 for approximately 12 months (Russell et al., 1996; English et al., 2013).
In the upper troposphere and lower stratosphere, the impact of each of these eruptions is distinct with a

near-complete recovery to background AOD levels after each event; however, at surface, a lag in recovery
time is evident (middle panel in Fig 5). The net impact of radiative flux perturbations following the
eruptions is summarized in the form of global surface air temperature change over the entire period. The
model produces a robust mean cooling of ~1.5º C on the second year after the first eruption, and although
AOD recovers a few years after each eruption, the surface temperature response is prolonged. The smaller

extratropical eruptions (E2-E4) that followed the large tropical one (E1) hindered the surface temperature
recovery and maintained a surface cooling of around 1.0ºC during the entire period of simulation. The





sea surface temperature (SST) response shown in fig S4 suggest a slow recovery of the oceans after the eruptions with a remanent cooling effect.

**3.3 Volcanic aerosol properties**

The effect of volcanic aerosols on radiative forcing is tightly controlled by aerosol size (Lacis et al., 1992; Hansen et al., 1980). The aerosol effective radius, $R_{eff}$, is a key metric in linking aerosol microphysical properties with their SW and LW impacts. The vertical profile of aerosol size as represented by $R_{eff}$ is calculated for each month and is shown in Fig 6. After the tropical eruption (E1), new aerosols nucleate

and grow rapidly via coagulation and, while $SO_2$ is still available, condensation, and attain a maximum $R_{eff}$ of greater than 0.5 $\mu$m approximately for 2 years. In comparison, $R_{eff}$ after the Pinatubo eruption went up to 0.6 $\mu$m and sustained that size for approximately 2 years (Russell et al., 1996). Sulfate aerosol sizes for the subsequent three smaller extratropical eruptions (E2 to E4) grow up to 0.3 $\mu$m.

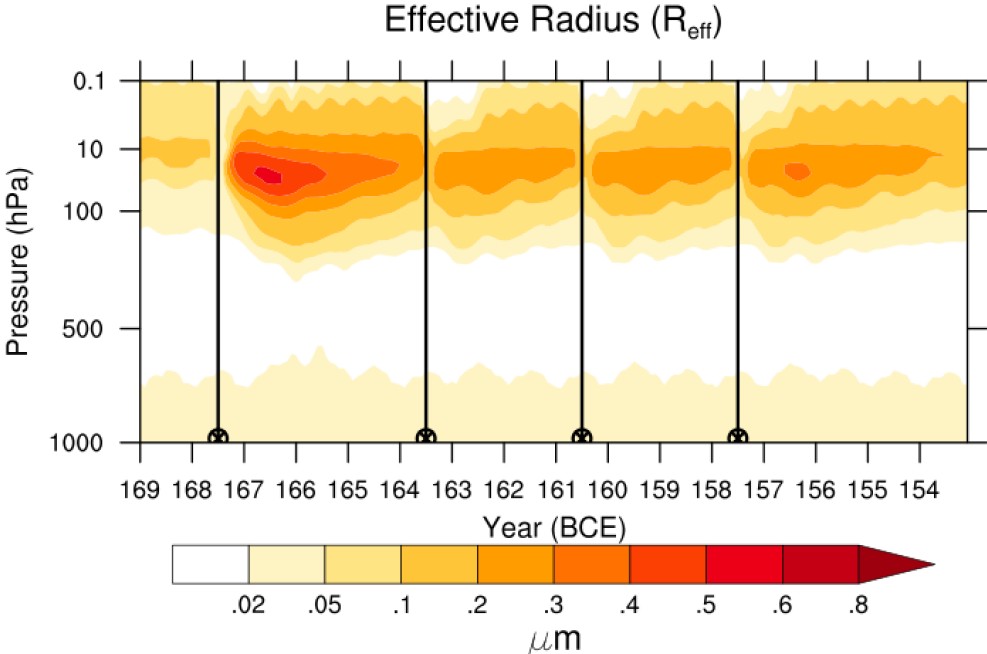

Fig 6.   Timeseries of the global ensemble mean vertical profile of sulfate aerosol $R_{eff}$ for the entire simulation period. The vertical black line with a circled cross mark at the bottom shows the timing of the eruptions.



The aerosol extinction vertical profile (Fig S5A) shows that the radiative impact of the E1 tropical

eruption in the lower stratosphere is prolonged as compared to the later extratropical eruptions. Heating

of the lower stratosphere affects the dynamics of the stratosphere; after tropical eruptions enhanced

tropical upwelling and extratropical downwelling with the phase of Brewer-Dobson circulation have an

impact on the transportation of trace species such as Ozone ($O_3$) and $NO_2$ (Aquila et al., 2013; Trepte et

al., 1992; Pitari et al., 2016; Pitari and Mancini, 2002). Fig S5B shows a strong positive ($\geq$10 ppbv)

anomaly of $CH_4$ in the upper stratosphere and negative ($\leq$10 ppbv) anomalies in the lower stratosphere,

especially after the tropical eruption (E1). Changes in the mean concentration of upper and lower

stratospheric methane ($CH_4$) suggest a strong vertical transport (Kilian et al., 2020).

### 3.4. Latitudinal temperature response to volcanic aerosol forcing

The Hovmöller diagram (Fig 7A and 7B) shows the difference of zonally averaged AOD at 550nm and

surface air temperature response between the ensemble mean of the volcanic eruption simulations

compared to the mean climatology of the control simulation. The pattern of total AOD after the first

eruption (E1) shows a strong cross-equatorial transportation of the stratospheric aerosols into the southern

hemisphere, with a similar pattern in the northern hemisphere. This is consistent with the hypothesis that

an enhanced Brewer-Dobson circulation in the southern hemisphere during the austral winter season led

to the southward transportation of volcanic aerosols after a Pinatubo type (tropical) eruption (Aquila et

al. 2012). The initial dispersal of aerosols from eruption E1 is strongly influenced by its timing and

exhibits a seasonal dependence (consistent with Toohey et al., 2011). However, the other three eruptions

(E2, E3 and E4) are in high latitude extratropics and only yield increased AOD in the northern hemisphere.

A lag of more than 12 months in the surface temperature response after the first eruption correlates well

with the distribution of aerosols, consistent with findings reported in the literature for similar events (e.g.,

Jungclaus et al., 2010; Klocke, 2011). The global mean surface temperature response peak appears when

the volcanic aerosols from the tropical eruption (E1) extend across the northern hemisphere extratropics

and the polar regions. It should be noted that the land surface over the northern extratropics responds

quickly to the attenuated post-eruption shortwave radiative flux compared to the tropics. The zonally

averaged surface temperature response (fig 7B) shows that a strong cooling of 1.0-1.5˚C lasts over the





tropical north and partially over the tropical southern hemisphere for more than 30 months after the first eruption. Further, the greater anomalies of >2.0 ˚C cooling mostly appear six months after the first eruption, with the subsequent extratropical eruptions helping to maintain the northern hemispheric cooling. The seasonality of surface temperature response reveals a more substantial cooling during the boreal summer season and also the expected winter warming pattern after tropical eruptions (E1) over Europe. Supplementary Fig S6 shows the spatial pattern of the surface temperature response to volcanic aerosols over the four seasons following the first eruption (E1) (JJA & SON for the year of eruption and DJF & MAM for the next year). The surface temperature response for the first two seasons is confined to the tropics and moves to higher latitudes after six months. As evident in fig S3, the anomalous winter (DJF) warming pattern after the eruption over Europe and the cooling over Northern America could be a product of the same fundamental atmospheric dynamics as noticed after Pinatubo eruption (Robock, 2000; Robock and Mao, 1992).



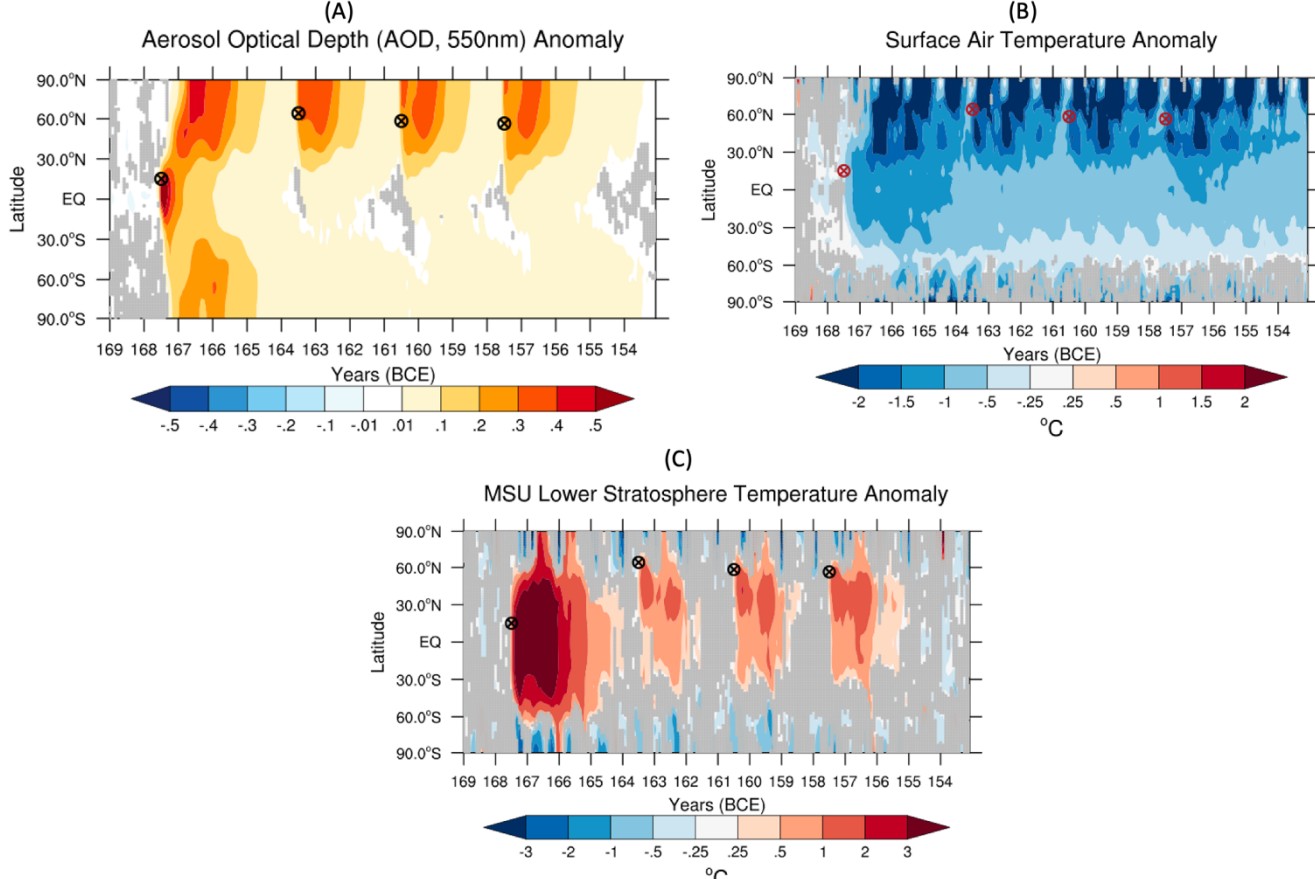

Fig 7. Hovmöller diagram showing the zonally-averaged temporal dispersion of volcanic aerosols in terms of AOD change at 550nm (A), surface temperature response (B), and lower stratospheric temperature response (C). Anomalies were calculated with respect to a climatological annual cycle calculated from the control simulation. The gray color is painted over the regions where changes are not statistically significant at the 95% confidence level. Circled cross marks show the modeled spatial and temporal

position of the eruptions.

The global lower stratospheric temperature response in terms of MSU TLS data is discussed in section 3.2. Interestingly, fig 7C shows that the latitudinal anomaly of the lower stratosphere warming is broadly limited to the equatorial lower stratosphere. Eruption E1 induces lower stratosphere warming on the order

of >3 °C, with a weaker warming of up to 1-2 °C after the three extratropical eruptions (E2, E3 and E4). Lower stratosphere warming also affects the polar vortex strength in the northern hemisphere and



atmospheric circulations into the troposphere, with substantial repercussions for surface climate and variability patterns as suggested in previous research (e.g., Graf et al., 1993, 2007; Shindell et al., 2004).

## 3.5 Latitudinal precipitation response to volcanic aerosols

Numerous studies using the observational record in addition to modeling efforts have demonstrated that the cascading impact of an altered radiative balance at the top of the atmosphere due to volcanic eruptions is reflected in the hydrological cycle in terms of regional patterns of seasonal rainfall change (Robock and Liu, 1994; Robock, 2000; Trenberth & Dai, 2006; Schneider et al., 2009, Iles et al., 2012; Iles and Hegerl, 2014; Timmreck, 2012). Societies conducting agriculture in arid and semi-arid regions before the advent of modern reservoirs and irrigation, such as in ancient Egypt, are perhaps most impacted by such changes. We thus investigate the hydrological cycle response to this set of eruptions at a global and regional scale, paying particular regard to the northern hemispherical monsoon season (JJAS) for the first 2 years following each eruption for the period under study. Fig 8 shows the Hovmöller diagram of the zonal mean precipitation anomaly relative to the annual cycle climatology of the 100-year-long control simulation.





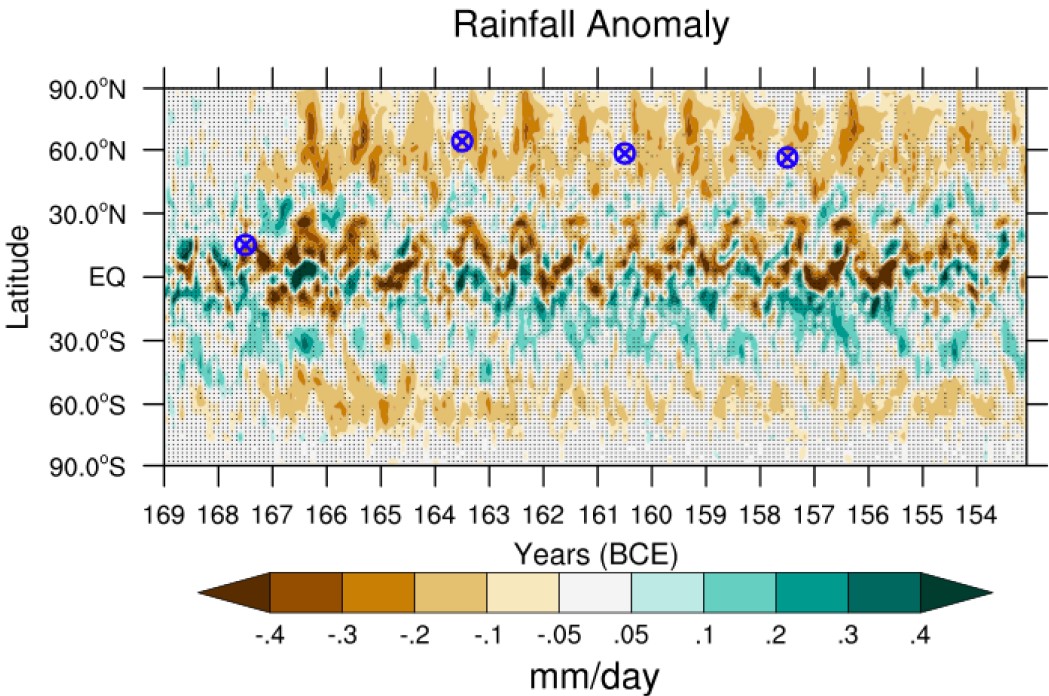

Fig 8. Hovmöller diagram showing the zonally averaged rainfall anomaly for the entire period as the
spatiotemporal response of global rainfall to our series of volcanic eruptions. Circled cross marks show
the locations and timing of the eruptions. Black dots point out the regions where changes are not
statistically significant at the 95% confidence level.

The ensemble zonal mean rainfall change after the eruptions shows a substantial negative trend in the

northern hemisphere as a result of cooling induced due to the volcanic aerosols. A robust negative

anomaly of the order of 0.3-0.4 mm/day in the northern hemisphere rain belt (ITCZ) region appears after

the first eruption and persists during the following years. A pattern of strong drying in the equator also

coincides with the northern hemisphere monsoon season (JJAS). However, rainfall response in the

northern hemisphere extratropics strongly correlates with the surface temperature response and thus

emerges here 12 months after the tropical eruption (E1), with the model calculating a moderate to a high

decrease on the order of 0.1-0.2 mm/day in rainfall, a response that persists throughout the year for three

post eruption years. A shift in the northern hemisphere rainfall pattern is also evident for the region around

30ºN, with slight increases in rainfall. This response is statistically significant over only a few spots after

the first eruption. Fig 8 clearly demonstrates that a drying pattern is also evident after the other eruptions



(E2-E4) and that because of this the northern hemisphere experiences a sustained net (albeit temporally varying) precipitation decline for the entire modeled period, and with a distinct seasonal character.

Fig 9. Mean change (mm/day) in northern hemisphere monsoon season (JJAS) rainfall averaged three consecutive years after eruption E1 and two years after each of E2, E3 and E4 (left to right and top to bottom). The caption over each panel shows the eruption characteristics. A gray color is painted over the grid boxes for which change in rainfall is not significant at the 95% confidence level. Years indicated in parentheses follow the order of the eruptions in our simulation period, i.e., E1 occurs in the 2nd year, and E2-E4 occur in the 6th, 9th and 12th years, respectively.

We further evaluated the spatial patterns of change in mean rainfall during the northern hemisphere monsoon season (JJAS) as shown in fig 9. We averaged the three monsoon seasons (eruption year and next 2 years) after the more potent tropical eruption (E1) and two monsoon seasons (eruption year and



next year) after each of the remaining extratropical eruptions (E2, E3 and E4), and focused principally on identifying statistically significant responses. Hence, after the tropical eruption, the summer monsoon rainfall appears strongly suppressed over many major northern hemisphere monsoon regions. Importantly for our historical focus on Egypt, African monsoon rainfall shows a notable decrease of 0.5-1.0 mm/day during the three-year post-eruption JJAS season average (which includes the eruption year). This decrease

covers a large area in Africa from (approximately) the equator to (approximately) 17°N. The South and East Asian monsoon regions are also shown to experience a robust negative rainfall anomaly of >1.0 mm/day over the Indian subcontinent as well as (more variably) several regions of China, though with some isolated increase over the eastern Vietnamese landmass. Similar patterns of decreased rainfall also appear over the western (and particularly northwestern) Pacific and northern hemispheric high latitude

regions more broadly. The model also simulated a (statistically significant) band of enhanced JJAS rainfall stretching from Central Asia westward through the Near East and into the Mediterranean, Western European and parts of the North Atlantic (roughly between a latitudinal band of 30°N to 50°N). A contiguous band of increased rainfall is also observed further south and west in the Atlantic, stretching into portions of the northern Caribbean, southeastern Gulf of Mexico and Mesoamerica (fig 9).

Similar patterns of suppressed monsoon season rainfall are observed following each of the extratropical eruptions (E2-E4), but a particularly notable east-west band over both land and ocean (broadly confined between slightly north of the equator and 30°S) shows a positive rainfall anomaly (being most clearly statistically significant between (approximately) 5°N and 10°S (fig 9)). This pattern is largely consistent with past literature employing observations and modeling of volcanic climatic impacts under a range of

scenarios and periods (e.g., Robock, 2000; Robock and Liu, 1994; Iles et al., 2012; Liu et al., 2016; Haywood et al., 2013; Schneider et al., 2009; Trenberth and Dai, 2007; Joseph and Zeng, 2011; Gu and Adler, 2011). In terms of mechanisms, for many northern hemisphere landmasses, these eruptions have clearly induced a surface cooling that has altered the northern hemisphere meridional (equator-to-pole) surface temperature gradient (fig 7). Given this energetic deficit, we may posit that the northern

hemisphere (NH) experienced a post-eruption alteration of large-scale circulation patterns and moisture convergence, resulting in a constrained northward migration of the ITCZ during the boreal summer, suppressing large scale rainfall patterns over many northern hemispheric monsoon regions and (as a





related consequence) promoting increased rainfall in the above-described band from the equator

southward (Liu et al., 2016; Oman et al., 2006; Graf, 1992; Dogar, 2018). This is consistent with
observations by Colose et al. (2016), who demonstrated that a hemispherically asymmetric volcanic
forcing creates energetically deficient conditions in the hemisphere of the greatest forcing and which
"pushes" the ITCZ away from it. It has also been shown with palaeoclimatic data that tropical and northern
hemisphere eruptions can create a dipole that results in wetter summer conditions over extensive parts of
the Mediterranean, with correspondingly drier conditions over northern Europe (Rao et al., 2017). This is

also largely consistent with our model output (fig 9).

### 3.6 African monsoon and Nile River response

Our modeling suggests that all four eruptions during the period 168-158 BCE are likely to have strongly
influenced the regional rainfall pattern over different monsoon rainfall regions in the northern hemisphere

consecutively for 2-3 years after each eruption, and in combination produced a sustained deficit in
monsoon rainfall (on average) for more than a decade. We now focus on the North African monsoon
region, which strongly affects the Nile River summer flooding. Fig 10 shows the 3 consecutive years of
monsoon season (JJAS) rainfall over equatorial and northern Africa (encompassing the Nile River basin)
after each eruption. The African monsoon shows a strong response with reduced rainfall of more than 1

mm/day following the (mid-June) E1 tropical eruption in the year of the eruption itself (Year 0, fig 10),
affecting both the White Nile watershed to the south of the basin and the Blue Nile and Atbara River
watersheds further north and east in the Ethiopian Highlands. Reduced precipitation is also observed
following each of the (also mid-June) extratropical eruptions (E2-E4) during the eruption years, but is
more spatially constrained (and for E2, less severe). This result is perhaps unsurprising as the $SO_2$ output

of E2-E4 is only approximately 1/3 that of the tropical eruption, E1. Nonetheless in each case the Blue
Nile and Atbara River headwaters in the Ethiopian highlands are observed to experience a statistically
significant decrease, with important implications for the summer flooding in Egypt, which depends for
approximately 80% of its floodwater on rainfall here (Melesse et al., 2011). For E2-E4, this response can
be seen to intensify in the first post-eruption year, persisting into the second post-eruption year, while for

E1 the response contracts geographically to resemble the response seen after E2-E4 (fig 9, fig 10). These



results are indicative of an effective suppression of the African monsoon following tropical and northern hemispheric extratropical volcanic eruptions, a finding consistent with previous studies (e.g., Colose et al., 2016; Oman et al., 2006; Haywood et al., 2013; Jacobson et al. 2020; Manning et al., 2017).

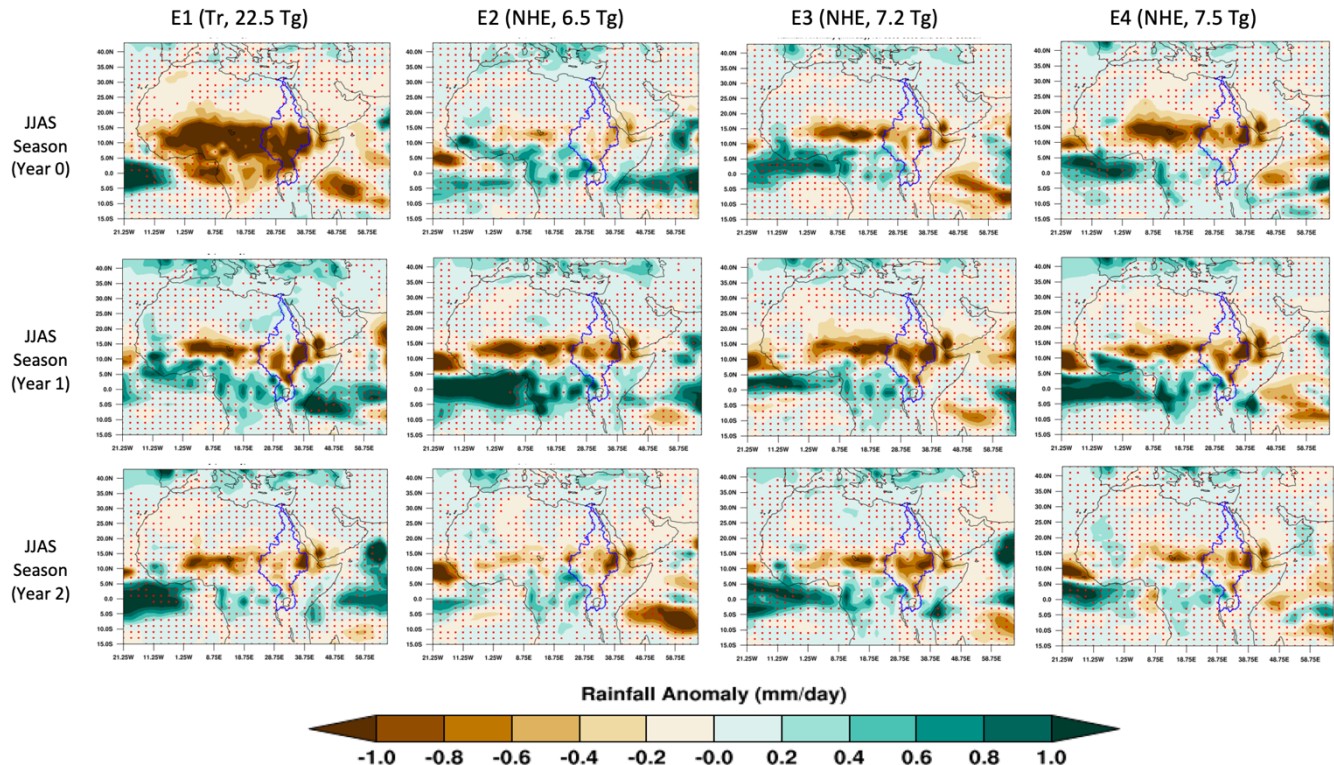


Fig 10. Ensemble mean rainfall difference from the climatological control for each of the first three monsoon seasons (JJAS; rows) after each eruption (columns) over equatorial and North Africa. The blue boundary line shows the present-day Nile River basin, which is broadly similar to the river extent approximately 2.5ka years ago. The red stippling indicates the regions over which change in rainfall is

not significant at a 95% confidence level.

Spatial patterns of total cloud cover (Fig S7) for the three consecutive post-eruption monsoon seasons show a cloud cover decrease of up to 10% over East Africa and the adjacent Indian Ocean region. These spatial patterns are consistent with the above-reported negative rainfall anomalies (>1 mm/day) over

North African land regions, especially over the watershed of the Nile River basin, and again suggest a strong weakening of the Indian summer monsoon (Graf, 1992, Oman et al., 2005). Positive anomalies of



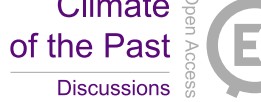

total cloud cover also coincide with regions observed as having a positive rainfall response (e.g., Mediterranean and Middle East).

We also analyzed the mass of total water flow averaged over the Nile River basin (blue line, fig 11) as
being representative of Nile flooding and river discharge at the river's mouth at an annual scale to summarize the volcanic impacts on Nile flooding. Table 2 thus presents the percentage deficit (-) or excess (+) of water flow in the Nile River basin on an annual basis after each eruption along with the variability (one standard deviation) observed across the model ensemble members, relative to the 100 years climatological mean (base climate). The tropical eruption (E1) has a strong impact (>30% deficit) on the
annual water mass over the Nile catchment during the eruption year and first full post-eruption year (i.e., Years 0 to 1, table 2), with a more moderate decrease of ~13% during the second full post-eruption year (i.e., Year 2, table 2). The first extratropical eruption (E1) shows a minor decrease in the eruption year (Year 0, table 2) but is not deemed statistically significant. The following two full post-eruption years reverse this pattern to exhibit a modest increase, but which is also not deemed statistically significant.
The next two extratropical eruptions (E3 and E4) instead show a more consistent response in the form of a decrease. For E3 and E4 this decrease is on the order of ~-5% in the eruption year (i.e., Year 0, table 2) and is notably greater in the first full post-eruption year (i.e., Year 1, table 2) for both E3 and E4 (being ~-18% and ~-12%, respectively). The decrease persists into the second full post-eruption year (i.e., Year 2, table 2) for E4 (~-12%) but effectively falls back in line with the 100 years climatological mean for E3
(although this annual whole-basin mean change does exhibit the highest observed variance among ensemble members (table 2)).

Table 2. Annual mean change (%) and standard deviation in water mass flow over the Nile River catchment for 3 consecutive years after each eruption.

| | E1(Tr, 22.5 Tg) Change /Std | E2(NHE, 6.5 Tg) Change /Std | E3(NHE, 7.2 Tg) Change /Std | E4(NHE, 7.5 Tg) Change /Std |
|---|---|---|---|---|
| Year 0 (eruption year, mid-June) | -28.7±39.9 | -3.02±22.5 | -4.9±35.2 | -4.7±29.6 |
| Year 1 | -37.8±22.5 | 2.5±36.7 | -18.1±28.9 | -11.7±29.9 |
| Year 2 | -13.4±32.2 | 10.7±39.9 | 0.9±47.8 | -12.1±28.0 |



The spatial patterning of response across a basin as complex as the Nile is a critical further consideration (fig 11). After the tropical eruption (E1), the previously-described rainfall suppression is associated with a drastic decrease in annual river flow over effectively the entire river basin, with a simulated decrease of approximately 30, 40 and 15 km$^3$ per year relative to the 100 years climatological mean (~104 km$^3$) for Years 0 to 2, respectively. After the second eruption (E2), the total annual river flow in Year 0 is observed

to slightly increase (table 2), although this response is not statistically significant, and in Year 1 exhibits a marked contrast between (particularly) the southern (greater flow) and northern (lesser flow) parts of the basin, before a more consistent increased flow is observed in Year 2. The contrast between a reduced flow over (broadly) the northern part of the basin versus increased flow over the southern part is then observed consistently for all post eruption years shown in fig 11 for both eruptions E3 and E4. We can

hypothesize that this contrast arises in large part as a function of the size and complexity of the Nile basin (and markedly different geographical location of rainfall supplying the White Nile to the south and Blue Nile and Atbara River to the northeast), combined with the asymmetrical loading of sulfate aerosols in the higher latitudes of the northern hemisphere after extratropical eruptions. This may lead to a post-eruption scenario in which the northward boreal summer migration of the ITCZ and associated rain-

bearing monsoon winds are suppressed (as discussed earlier). Given that these winds are the primary driver of summer rainfall over the Ethiopian highlands, the summer flooding of the Blue Nile and Atbara River into the north of the basin and Egypt will be diminished, while water flow down the White Nile (fed by rainfall over the equatorial lakes) will be enhanced by the failure of the ITCZ to migrate northward beyond this region.






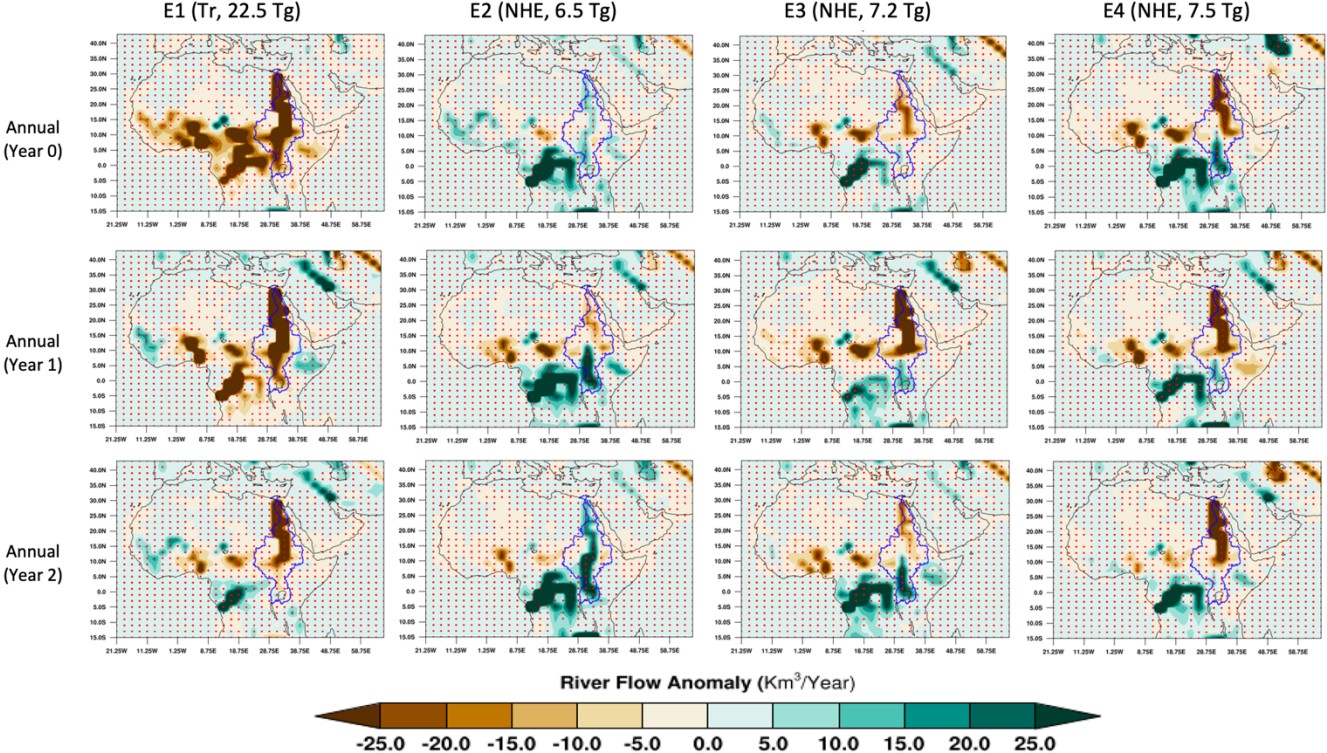

Fig 11. Annual river flow anomaly (km³/year) from the control climatology for 3 consecutive years after each eruption (columns) over the North African continent. Other details are same as Fig 10.

To summarize the story of the hydroclimatic impact of these four sequential volcanic eruptions on the Nile River basin, Fig 12 (top panel) shows that the northern hemisphere experienced a substantial cooling of ~2.5° C (1.0° C greater than the global average response) with a lower spread among ensembles after the first eruption (E1). The subsequent eruptions (E2, E3, and E4), occurring at equal temporal intervals, then maintained a cooling of ~1.75°C for at least a decade. The monthly anomaly of mean rainfall over

the Nile basin is observed as a considerable decrease of ~1.0 to 1.5 mm/day during the monsoon season (JJAS) for the years following each eruption (Figure 12, middle panel). The impact of decreased rainfall over this region is strongly evident after the tropical eruption E1 in terms of Nile River discharge at the river mouth (grid box centered at 29.0N, 31.25E) in the Nile delta region of Egypt (Fig 12, bottom panel). Our modelling shows here a mean deficit that begins in the year of the eruption (i.e., Year 0) and peaks

at a reduction of more than 50% of water discharge during the first full post-eruption year (i.e., Year 1) that takes effectively 2 further years to recover. There is no persistent negative discharge anomaly evident



after the second eruption (E2), although variability around the mean is quite high here. By contrast, a deficit is observed to begin in the year of the third eruption (E3, i.e. Year 0) that then persists throughout the first full post-eruption year (i.e., Year 1) and into the start of the second full post-eruption year (i.e., Year 2). This deficit peaks in Year 2 at just less than 50%. A similar response is observed after the fourth (E4) eruption, with a persistent anomaly starting in the first full post-eruption year (i.e., Year 1), continuing throughout Year 2 and into the start of Year 3. This deficit also peaks in Year 1 at approximately 30%.

Fig 12: Monthly time series of surface temperature response (˚C) averaged northern hemisphere (NH) (top panel), mean rainfall change (mm/day) over the spatial box over Nile River watershed (Latitude:





(5N, 18N), Longitude: (30E, 42E)) (middle panel) and Nile River discharge anomaly (%) at the delta
region (grid box centered at 29.0N, 31.25E). The dark solid line shows the multi-ensemble mean, and the
color envelope shows the associated variability ((±σ; Standard deviation). The vertical dotted green line
shows when each eruption happens.

## 4. Discussion and Conclusions

Recent years have seen increasing interest in the role of hydroclimatic variability in human history,
including by interdisciplinary teams combining evidence and methods from across traditional disciplinary
divides (e.g., McCormick, 2011, 2019; Manning et al., 2017; Ludlow and Travis, 2019; van Bavel et al.,
2019; Degroot et al., 2021; Ljungqvist et al., 2021; Izdebski et al., in-press). For the pre-modern era, when
systematic observations of hydroclimate become scarce, this effort depends increasingly upon natural
archives (palaeoclimatic proxies) that track variability at spatial and temporal resolutions sufficiently
high, and boasting sufficiently accurate dating, to convincingly identify associations with societal
phenomena (e.g., subsistence crises, migration, conflict), economic and demographic processes, and
major historical events (e.g., the "collapse" of kingdoms and empires). Work such as that by PAGES2k
Network members offering paleoclimatic reconstructions and curated data collections (e.g., PAGES 2k
Consortium, 2013; PAGES 2k Consortium, 2017) are thus crucial, although here the exclusive focus on
the past 2k years (for some proxies an artificial horizon and for others an aspiration as temporal coverage
and sampling depth improves), presently excludes some of the most foundational periods and events in
human history. This includes the development of advanced ancient societies in Asia, the Near East and
Mediterranean that are richly documented and hence offer considerable potential for the study of
socioecological systems.

Important work has still been possible using speleothems, sedimentary and other archives (e.g., Drake,
2012; Schneider and Adali, 2014; Knapp and Manning, 2016; Sołtysiak, 2016), but there is often little
direct temporal and/or geographical overlap between these early ancient world regions of rich human
documentation and proxies (e.g., tree-ring based) with precision and accuracy at annual-or-better
resolutions. A major recent advance, however, has been the publication of a chronologically precise and
accurate bipolar ice-core-based volcanic forcing reconstruction for the past 2,500 years (Sigl et al., 2015;



Toohey and Sigl, 2017). The potentially global hydroclimatic impacts of major explosive eruptions makes this record widely geographically relevant, while the repeated incidence of major eruptions that can now be seen in ever greater detail through sulphate deposition in the polar ice sheets has allowed their use as
"tests" of societal vulnerability and response to sudden hydroclimatic shocks in both a statistical manner (e.g., Manning et al., 2017; Gao et al., 2021; Ludlow et al., in-press, 2022) and in a complementary qualitative manner as "revelatory crises" (Solway, 1994; Dove, 2014), in which tensions, inequalities and vulnerabilities in given political and economic systems are potentially exposed under the pressure of sudden environmental variability (e.g., Ludlow and Crampsie, 2019; Ludlow et al., in-press, 2022). The
scrutiny that such exposure can bring to prevailing human systems (social, economic, ideological) can spur their potentially rapid transformation, while the related suspension of cultural norms and business-as-usual practices under the "state of exception" that may prevail during "natural" disasters and other crises can facilitate this, though such conditions can also be exploited to further entrench existing power bases (e.g., Dove, 2014; McConnell et al., 2020).

For historical eruptions to act as tests or be studied as potential "revelatory" crises, knowledge of their dating alone is insufficient, particularly given the regional and seasonal variability of volcanic hydroclimatic impacts, and the sensitivity of these impacts to multiple variables such as the location, season, chemical composition and height attained by volcanic ejecta (Robock, 2000; Cole-Dai, 2010; Ludlow et al., 2013). Even where instrumental or natural archives are available, but especially where
these are thin or absent, climate modelling can provide insights into the expected climatic impacts for particular regions, seasons and related physical (e.g., riverine) systems. This is true for modelling of idealized eruptions, but potentially even more so for models that produce "historical realizations" based upon actual forcing reconstructions (e.g., Tardif et al., 2019). In this context we have presented a modeling effort that explores the impacts of a unique eruption quartet during the (historically tumultuous) decade
168-158 BCE, with a particular focus on the Nile River basin. These target years are intermediate between the mid-Holocene and end of the preindustrial periods, and representative background climate conditions are necessary to investigate the climatic impact of such a short-term forcing (Zanchettin et al., 2013). PMIP4 vegetation distributions (linearly interpolated for the 2.5ka period from the mid-Holocene (Otto-Bliesner et al., 2017) to the end of the preindustrial) for the GISS ModelE2.1 (MATRIX) version (Kelley



et al., 2020; Bauer et al., 2020) were therefore used to improve GCM simulations without a fully dynamic vegetation implementation (Harrison et al., 2015). Vegetation-albedo feedbacks due to a greater prevalence of arid shrubs/steppe over Africa and of boreal forests over high latitudes were thus observed to induce a northward movement of the ITCZ over Africa (Sahara region) promoting a simulated rainfall increase of the order of 0.5-1.0 mm/day in the region (such a response is consistent with theoretical

expectations and other estimates (e.g., Otterman 1975; Charney 1975; Claussen 2009; Pausata et al., 2016; Rachmayini et al., 2015)).

The GISS ModelE2.1 simulated a strong shortwave and longwave global radiative forcing of -10 and +3.0 W/m$^2$, respectively, following the tropical eruption (E1) and a roughly equal forcing of -3.5 and +1.0 W/m$^2$, respectively, for each of the 3 extratropical eruptions (E2-E4). The peak net radiative volcanic

forcing was calculated at -7.5 W/m$^2$ and 2.5 W/m$^2$ for the tropical and extratropical eruptions, respectively. The model calculated a global AOD at 550 nm of 0.22 and 0.1 after the tropical and extratropical eruptions, respectively, and estimated a peak cooling of ~1.5 ºC almost 12-months after the first eruption (E1), with the three consecutive eruptions then sustaining a surface cooling of about 1.0 ºC for almost all of the 15 years of simulations. The first eruption (E1) was 30% larger than Pinatubo and

the GISS ModelE2.1 simulated proportionally stronger radiative impacts as compared to Pinatubo (for details of which, see: Hansen et al., 1992; Robock and Mao 1994; parker et al., 1996; McCormick et al., 1995; Stenchikov, 2015). A detailed analysis of the impacts of volcanic aerosols on the chemical composition of the stratosphere was not part of this study.

The global hydrological cycle responds vigorously to the volcanically-induced surface cooling in the

GISS ModelE2.1, with a greater than 1.0 mm/day decrease in rainfall over the African, Indian, and Chinese regions during the summer monsoon season consecutively for 3 years after eruption E1 (tropical) and for 2 years after each of the eruptions E2-E4 (extratropical northern hemispheric). Statistically significant decreases in rainfall over the major tropical northern hemisphere rain belt is also calculated by the model, as well as more broadly over higher latitudes for this hemisphere. Some smaller regions of

positive rainfall anomalies are, however, simulated over the northern hemisphere mid-latitudes (both land and ocean) around 30º N. These patterns of hydrological cycle response are consistent with previous studies reporting changes in rainfall and large-scale atmospheric circulations (such as Hadley cell





weakening) (e.g., Robock and Liu, 1994; Gillett et al., 2004; Trenberth and Dai, 2007; Crowley et al., 2008; Fischer et al., 2008; Joseph and Zeng, 2011; Timmreck, 2012; Iles et al., 2012, Haywood et al., 2013; Liu et al., 2016).

For the equatorial and northern African landmass specifically, the GISS ModelE2.1 produces a notable suppression of monsoon (JJAS) rainfall for all eruptions, E1-E4. The onset of this response can be observed in the JJAS season beginning with each eruption year itself, though the timing of the peak intensity and/or greatest spatial extent of this suppression can vary between eruptions (e.g., for E1 the greatest extent and peak intensity occurs for JJAS in Year 0, while for E2-E3 the peak intensity and greatest extent occurs in Year 1, and for E4 in Year 0). The suppression centers (for all eruptions and each plotted post-eruption JJAS season, fig 10) around latitudes 10-15°N, where it runs in an east-west band that in some years is effectively contiguous across the continent (approx. 16°W to 52°E). There is, however, a tendency for this response to be more marked and long-lived (into JJAS of Year 2, fig 10) in the central and eastern portions of this range, where it is statistically significant and can surpass 1 mm/day (up to 30-40% of climatology for control period).

Importantly, the regions of the most rapid onset, greatest persistence and intensity of response include Lake Tana (12°0′N 37°15′E) and the Ethiopian highlands that comprise the headwaters of the Blue Nile and Atbara rivers and supply the vast majority of summer floodwater in Egypt (Melesse et al., 2011). This result is broadly consistent with CMIP5 model runs forced with large twentieth century eruptions (e.g., Iles and Hegerl, 2014; Manning et al., 2017). Annual river flow for the Nile River basin (fig 11) closely follows the apparent patterns of decreased JJAS rainfall over the headwater region. Simulated river flow shows a deficit in the range of 15-40 km$^3$/year up to 3 years following the modelled extratropical northern hemisphere eruptions. Simulated variability in river discharge is also seen to increase 3/4-fold following the extratropical eruptions, because of the spatial variability in the rainfall response across ensemble members. There is no way to tell which ensemble member describes best the historical conditions that actually happened following the eruptions between 168 and 158 BCE, but the large variability and the statistical significance of the drying tells us that Nile summer flooding may have been considerably lower than the simulated mean anomaly.





What is nonetheless certain is that the scale and persistence of the hydroclimatic impacts implied by our modelling for the 168-158 BCE eruption quartet supports, to begin, inferences of poor Nile flooding in 166 and 161 BCE from scattered references in surviving written sources (Bonneau, 1971). These also identify 169 BCE as potentially experiencing poor flooding, which suggests (assuming sufficiently accurate ice-core dating (Sigl et al., 2015)) that the eruption quartet may have compounded any societal

impacts already arising from this. Indeed, our modelling supports the contention that there is a largely overlooked but significant environmental context to what has long been recognized as a tumultuous decade in Egyptian history. Acknowledging the historical context evolving over the preceding decades, and the resulting political, military, economic and cultural setting through which any hydroclimatic shock will have propagated, is of course also key to achieving a fuller understanding of the human-

environmental entanglements in the 160s and thereafter, including the role of explosive volcanism. Thus, the increasing dominance of Rome in the eastern Mediterranean, and the growing internal political weakness of the Ptolemaic kingdom and their great rivals the Seleukid empire are writ large in the historical narrative of the second century BCE Mediterranean world, but the extent to which the instability of the major eastern Mediterranean powers was an outcome of the rising power of Rome has been heavily

debated. A consensus view is now that these developments were directly coupled, and that eastward Roman expansion was driven not by the exceptional aggressiveness of Rome so much as by a "power transition crisis" in the eastern states around 207-200 BCE that drew Rome in (Eckstein, 2008).

The Ptolemies also began to face notable internal dynastic disputes and broader internal unrest among (at least certain sections of) the populace in the years after the Battle of Raphia in 217 BCE. Despite their

success in that battle against the Seleukids at the close of the so-called Fourth Syrian War (Grainger, 2010), this was marked as a turning point for longer-term Ptolemaic fortunes by the Greek historian Polybius (V.107.1-3). The high cost of this war and subsequent continued conflict with the Seleukids (that also ultimately saw the Ptolemies lose control of important rain-fed agricultural regions such as Coele Syria that had given the kingdom some resilience to years of poor Nile flood), were important

developments that likely increased their vulnerability to volcanic hydroclimatic shocks. Indeed, other eruptions such as a tropical eruption in 209 BCE (Sigl et al., 2015) occur conspicuously close to other major events such as the Great Theban Revolt that started ca.207 BCE (Manning et al., 2017), in which




the Ptolemies lost control of Upper (i.e., southern) Egypt to two presumably native Egyptian kings until
187 BCE, with unrest also extending at times into the Delta region of Lower (i.e., northern) Egypt. During
the 6th Syrian War, Antiochus IV and his Seleukid army invaded Egypt twice. The first invasion occurred
in 170 BCE and the second, more serious occupation, occurred in 168 BCE. This takeover would have
re-shaped Mediterranean history had it not been averted by Roman diplomatic intervention commonly
referred to as "the Day of Eleusis" (Hölbl, 2001, pp.147-8).

It is against the background of these longer-term developments, to which explosive volcanism and
hydroclimatic shocks also likely contributed (Ludlow and Manning, 2016; Manning et al., 2017; Ludlow
and Manning, 2021), that our modelling allows us to more readily understand the internal turmoil in Egypt
in the 160s and 150s BCE, affecting both the capital Alexandria and the countryside. Surviving sources
refer, for example, to "bad times and been driven to every extremity owing to the price of wheat" in 168
BCE (*UPZ* 1 59; Bagnall and Derow, 2004, pp. 281-82), and it is known that by the middle of the decade
an Egypt-wide agricultural crisis, described as a "calamity" was underway that drove Ptolemaic officials
to near panic (*UPZ* 1 110, 165-164) BCE. Manning et al. (2017) have already identified dates of probable
revolt onset in Ptolemaic history, with such onset dates identified in 168 BCE and 156 BCE both also
coinciding closely with the dates of our eruption quartet. A study of the longevity and geography of these
revolts is now of considerable interest. The surviving texts do not yet tell a complete story, but scattered
written references that suggest a long persistence of revolt throughout the decade, including for the years
168-157 BCE (Veïsse 2004, pp. 78-79), are now rendered more credible and explicable given the
modelled persistence of reduced temperatures and suppressed Nile summer flooding for more than a
decade following the 168 BCE tropical eruption and the three following extratropical NH eruptions.



**Code/Data availability**

Details to support the results in the manuscript is available as supplementary information is provided with the manuscript. Raw data and codes are available on request to author.

**Acknowledgements**

RS, KT, FL and JM acknowledge support by the National Science Foundation under Grant No. ICER-1824770. ANL acknowledges institutional support from NASA GISS. Resources supporting this work were provided by the NASA High-End Computing (HEC) Program through the NASA Center for Climate Simulation (NCCS) at Goddard Space Flight Center. The authors thank for their input through multiple

discussions the project members and collaborators of the ICER-1824770 project, 'Volcanism, Hydrology and Social Conflict: Lessons from Hellenistic and Roman-Era Egypt and Mesopotamia'. FL acknowledges support from the Trinity Center for Environmental Humanities. This paper benefited from discussion facilitated by the 'Volcanic Impacts on Climate and Society' (VICS) Working Group of PAGES.


**Author's contributions**

FL and JM identified the study period in consultation with the other authors. RS, KT and ANL designed the model simulations. RS performed the simulations, created the figures in close collaboration with KT, ANL, FL and JM. RS wrote the first draft of the manuscript and led the writing of subsequent drafts. All

authors contributed to the interpretation of results and the drafting of the text.

**Competing interests**

The authors declare no competing interests.

**Short Summary**

This study is a modelling effort to investigate hydroclimate impacts for the Nile River basin induced by a volcanic "quartet" of four closely spaced eruptions in ice-core volcanic chronology for the decade 168-158 BCE in a context to ancient Egyptian history. The NASA GISS ModelE simulated a strong response



in sustained temperature reduction and suppressed monsoon rainfall over East Africa following these
eruptions, leading to a deficit in Egypt's agriculturally critical Nile summer flooding.

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
