# Peer review of "Investigating hydroclimatic impacts of the 168-158 BCE volcanic quartet and their relevance to the Nile River basin and Egyptian history"

_Climate of the Past, 2022_

## Referee Comment (RC1)

This article employs paleoclimate modelling to investigate the impacts of volcanic eruptions on hydroclimate, particularly the African monsoon and Nile flow, and thereby to assess whether and how historical eruptions may have been responsible for revolts in Ptolemaic Egypt. The study represents a valuable step in integrating historical research and paleoclimate modelling. However, the article could benefit from substantial reorganization, and it requires a clearer discussion of whether and how to attribute historical societal impacts to volcanic eruptions and climatic variability.

I would recommend substantially reducing and reorganizing the introduction for greater precision, clarity, and a logical flow. Currently, this section is very long and shifts among a number of topics. The introduction needs to establish only the following contexts and in the following order: (1) Volcanic eruptions are a major driver of historical climatic variability. (2) This includes suppression of precipitation, the ITCZ, and the African monsoon. (3) Thus, volcanic eruptions probably reduced the flow of the Nile. (4) Nile flood levels were historically crucial for Egyptian agriculture and thus the populations and states that relied on that agriculture. (5) There is a correlation between the timing of volcanic eruptions and timing of revolts in Ptolemaic Egypt but not sufficient historical records to demonstrate that there was a low Nile flow during those years. (6) Therefore, this study uses paleoclimate modelling to determine to what extent volcanic eruptions such as those experienced in the Ptolemaic period were sufficient to suppress the flow of the Nile. (7) This study can enhance our understanding of volcanic forcing of the climate, as well as the study of Egyptian history and the integration of paleoclimate and historical research.

Most of the other material currently in the introduction, including the discussion of climate as a causal factor in Egyptian history, should be edited out or moved to the discussion section.

The introduction should also acknowledge previous research on volcanic eruptions, Nile flood levels, and famines in during recent centuries, for which there are Nilometer measurements and abundant historical records—see especially Alan Mikhail, 'Ottoman Iceland: A Climate History,' *Environmental History* 20 (2015): 262–84. https://doi.org/10.1093/envhis/emv006. This research, particularly for the Ottoman era, already makes a strong case that volcanic eruptions have had major historical impacts on Egyptian society by causing low Nile flow, shortages, epidemics, and political instability (indeed, a stronger case, with richer detail, than is possible for ancient history). The real question is whether this was also the case in Ptolemaic period.

The article's arguments regarding attribution of societal impacts to volcanic eruptions are often imprecise. I would stress that the attribution of societal impacts to climate variability should be as clear and logical as the attribution of climate impacts to climatic forcings. In this case, the authors aim to evaluate whether and to what extent volcanic eruptions were responsible for revolts in Ptolemaic Egypt. They have made a *prima facie* case for a causal connection in previous research, which demonstrated a correlation between the timing of eruptions and timing of revolts. Now they are taking this causal argument one step further.

In this regard, the article should first specify its causal argument(s), preferably in contrastive terms. (For more on this issue, see e.g., S. White and Q. Pei. 'Attribution of Historical Societal Impacts and Adaptations to Climate and Extreme Events: Integrating Quantitative and Qualitative Perspectives'. Past Global Changes Magazine 28, no. 2 (2020): 44–45. https://doi.org/10.22498/pages.28.2.44 ) Do the authors mean to argue that the *presence* (rather than absence) of volcanic eruptions caused the *occurrence* (rather than nonoccurrence) of revolts?  Or do they mean to argue that the *timing* of the volcanic eruptions explains the *timing* of the revolts (which may have occurred anyway but in different years)?  Or is it some other distinction about the eruptions or climate forcing that explains some other difference in societal impacts?  I would stress that these are each very different arguments (though not mutually exclusive).  They each require different evidence and each have different implications for Egyptian history.  Until the authors specify which causal argument(s) they are making, it is difficult to determine whether they have succeeded or failed.

If the article intends to determine whether and to what extent the *occurrence* of eruptions were responsible for the *occurrence* of revolts in Egypt, then that will require a more clear and rigorous approach to causation.  To clarify this problem, and to avoid some of the confusion that often clouds discussions of climate impacts on human societies, it may help to use a simple analogy.  Let us suppose a doctor prescribes vicodin (v) to a bus driver without offering appropriate warnings about its side effects.  The bus driver subsequently causes a road accident in which another driver is injured.  The injured party sues the doctor on the basis that the negligent prescription (v) caused erratic driving (d) and therefore the accident (a) and the injury (i).  In common law, to demonstrate the doctor's responsibility for the injury the injured party would have to demonstrate with a preponderance of evidence at least the following two points: First, that the negligent prescription for vicodin was specifically necessary for the injury to occur (i.e., the "but-for" test).  Second, that negligently prescribing medication is somewhat sufficient to cause injuries in general (i.e., the "harm within risk" standard).  We could also express these two causal chains as two sets of conditional probabilities that would have to meet a reasonable threshold: first, $p(v|d)$, $p(d|a)$, $p(a|i)$ and second, $p(D|V)$, $p(A|D)$, $p(I|A)$, where lowercase letters stand for specific real-world events and the capital letters stand for a type of event in general.  These legal standards capture everyday understandings of causation and responsibility as well as centuries of philosophical discussion and legal experience.

While all this might seem a long way from volcanoes and instability in Ptolemaic Egypt, the issue of attribution here is basically the same.  To what extent was a volcanic eruption (v) responsible for political instability (i), throughout the mechanisms of drought (d) and famine (a)?  To attribute the political instability to the eruption, a preponderance of evidence should demonstrate a strong chain of specific necessity and at a least a weak chain of general sufficiency from (v) to (d) to (a) to (i).  If there were alternative sufficient causes and the eruption was not necessary for the outcome—let's say another climatic event would have caused a drought even in the absence of an eruption—then we cannot attribute the societal impact to the volcano at all.  If the chain of causation depended on extraordinary contributory factors—let's say the Ptolemaic empire was unusually reckless or vulnerable to instability (not wearing its seatbelt, metaphorically speaking)—then the causal responsibility of the eruption would be much diminished, and it would be misleading to refer to the eruption, rather than weaknesses within the empire, as "the cause" or even "a cause" of the occurrence of revolts.  Much of the historical discussion in the paper suggests this may have been the case.

What this study has done is to take a one small but important step toward demonstrating potential causal responsibility of volcanic eruptions for Egyptian instability by demonstrating the causal sufficiency of eruptions for Nile droughts in general: $p(D|V)$.  The paper needs to put this contribution in perspective and not claim to do either more or less.  It should neither

hide nor exaggerate the significance of this contribution with vague language about volcanoes "playing a role" or an "environmental context" for the disaster. It is entirely possible that we could one day demonstrate that volcanoes were causally responsible for revolts in Egypt, with similar standards and rigor that courts use to assign legal responsibility for damages. This is more than "playing a role": it is causal responsibility. However, this would require further research into other steps in those causal chains, including comparisons with better documented episodes during the medieval and Ottoman eras. On the other hand, if there were alternative sufficient causes of the drought, famine, or instability, or if Ptolemaic Egypt only faced problems because it was extraordinarily vulnerable, then it does not make sense to talk about the eruption as the cause of revolts at all (except perhaps as a trigger for the *timing* of the revolts). Talk about "a role" for the eruptions would be more misleading than helpful.

Nor does it help to include additional historical context (i.e., lines 795-843) if that context is not clearly addressed to a causal argument. If the authors intend to state that there were (or were not) alternative sufficient causes for Egyptian revolts besides eruption-induced droughts, then they should state that clearly. If they intend to state that changes in Egyptian leadership explain why some eruptions were followed by revolts but other were not, then they should also state that clearly. Otherwise, readers are left to infer causal arguments where the authors may not have intended them and where they may not be warranted. I can see that the authors are aiming for greater subtlety and sophistication; however, additional information that is not clearly tied to the causal argument(s) creates more confusion than clarity. Clearly, this study cannot yet provide a definite answer to the question of causal responsibility of volcanic eruptions for the occurrence (or is it timing?) of Egyptian revolts—nor does it need to. However, the authors need to be clear what contributions they *can* make to this question: that is, how we may update our assessments of the probabilities of necessity and sufficiency along relevant chains of causation. They may also explain what questions remain to be answered and how further research might address them.

The sections on climate modelling are mostly beyond my area of expertise to evaluate. However, with respect to evaluating historical societal impacts, I would question the emphasis on mean precipitation anomalies. To evaluate whether eruptions were a sufficient cause of a low Nile flow, what I really want to know is how much more probable a low Nile flow would be with an eruption vs. without an eruption: $p(D|V)/p(D|\neg V)$. That is, I need some help in assessing the counterfactual scenario: if there hadn't been those eruptions, would there probably have been droughts in Ptolemaic Egypt anyway? The conclusion on lines 578-580 ("likely to have strongly influenced") is too vague. The crucial issue in attributing societal impacts to volcanoes is just how likely it was that deficient Nile flows occurred due to eruptions.

Much of the material currently in the introduction and results sections reads more like discussion. I would encourage the authors to create a larger discussion section in two parts: one for the discussion of volcanic forcing and hydroclimate anomalies and another for discussion of societal impacts. The article would also benefit from a real conclusion that summarizes findings and returns to issues raised in the introduction. The authors may also wish to address the methodological significance of the work and, in particular, make proposals for further integration of paleoclimatology, climate modelling, and human history.

Specific issues:

Line 15: The phrase "sometimes widespread" is confusing. Based on context, I would suggest "both local protests and widespread revolts".

Line 24: I assume that "observe" here refers to finding an average in the simulations, not an actual observation of the real climate. Please clarify.

Line 55: This statement already presupposes the conclusion.

Line 56: The phrase "potentially climatically effective" is awkward. I would recommend perhaps "eruptions that may have had regional or global climatic impacts."

Line 57-58: Again, this statement presupposes the conclusion.

Line 146-152: I do not find that this example supports the authors' arguments. Instead, it serves as a reminder that there were, at times, other sufficient causes of political change in Egypt besides climatic variability, such as conflicts with neighbouring empires.

---

## Author Comment (AC1)

*Draft Reviewer Responses*
**Ram Singh et al. - Investigating hydroclimatic impacts of the 168–158 BCE volcanic quartet and their relevance to the Nile River basin and Egyptian history**

**Reviewer 1**
**1.0. This article employs paleoclimate modelling to investigate the impacts of volcanic eruptions on hydroclimate, particularly the African monsoon and Nile flow, and thereby to assess whether and how historical eruptions may have been responsible for revolts in Ptolemaic Egypt. The study represents a valuable step in integrating historical research and paleoclimate modelling. However, the article could benefit from substantial reorganization, and it requires a clearer discussion of whether and how to attribute historical societal impacts to volcanic eruptions and climatic variability.**

**I would recommend substantially reducing and reorganizing the introduction for greater precision, clarity, and a logical flow. Currently, this section is very long and shifts among a number of topics. The introduction needs to establish only the following contexts and in the following order: (1) Volcanic eruptions are a major driver of historical climatic variability. (2) This includes suppression of precipitation, the ITCZ, and the African monsoon. (3) Thus, volcanic eruptions probably reduced the flow of the Nile. (4) Nile flood levels were historically crucial for Egyptian agriculture and thus the populations and states that relied on that agriculture. (5) There is a correlation between the timing of volcanic eruptions and timing of revolts in Ptolemaic Egypt but not sufficient historical records to demonstrate that there was a low Nile flow during those years. (6) Therefore, this study uses paleoclimate modelling to determine to what extent volcanic eruptions such as those experienced in the Ptolemaic period were sufficient to suppress the flow of the Nile. (7) This study can enhance our understanding of volcanic forcing of the climate, as well as the study of Egyptian history and the integration of paleoclimate and historical research.**

We thank the reviewer for their obviously considerable time and effort in reviewing this paper in such detail, for the frank assessment of its merits and limitations, and constructive guidance on possible ways forward. We agree, to begin, that the article's overall Introduction should be amended and more extraneous material condensed or removed. In our revisions, we have thus removed several paragraphs outright, which are now accessible in the Supplement. We have also emphasized the key points recommended by the reviewer. In our revisions to the introduction (for more on which see our responses to the specific points raised by the reviewer below), we have also clarified the intent and scope of this article which, in our initially submitted draft, was clearly not sufficiently conveyed.

The article itself has been intent primarily on (1) detailing the efforts (methodological) to credibly model the hydroclimatic impacts (particularly for the Nile basin) of a set of four closely spaced explosive eruptions as registered in polar ice-cores between 168 and 158 BCE, for a period sufficiently remote in time as to require a careful accounting for

model parameters such as vegetation cover that was at the time of these eruptions meaningfully different from the modern era. The intent following from this was then (2) to present the results of this modeling in the context of an ongoing interdisciplinary project (US NSF Award #1824770: "Volcanism, Hydrology and Social Conflict: Lessons from Hellenistic and Roman-Era Egypt and Mesopotamia") to more broadly establish the socioeconomic and cultural impacts of hydroclimatic variability arising from historical volcanism during the Ptolemaic period in Egypt (305-30 BCE).

Given the space required to do justice to the modeling efforts and results, the goal of the paper is therefore not to break new ground, per se, in assessing the societal impacts of the "volcanic quartet" of 168-158 BCE. Rather, the goal is to present the modeling results here in full, providing a (modeling) foundation for carrying out such an assessment in a later (informal follow-up) paper, without competition here for space with the many other relevant lines of historical and archaeological evidence that are being considered in this follow-up paper.

In presenting the modeling results here, we do wish however to (1) reflect upon (as per our Discussion section) the importance of modeling as a contributor to interdisciplinary studies of human-environmental entanglements, (2), present the model results in the context of the project's work to date on establishing a diachronic statistical link between political activity such as revolt and volcanically induced hydroclimatic variability in Ptolemaic Egypt (as per Manning et al. (2017)), and (3), set the stage for how a close case study of a particular decade known for its political instability and (now) for its likely marked hydroclimatic stress in Egypt (i.e., the 160s BCE), can allow us to push further in understanding underlying causal linkages.

In the above respect, the reviewer's methodological guidance on causality is certainly relevant to highlight, though its actual practical application in the present paper is beyond its intended scope.

**1.1. Most of the other material currently in the introduction, including the discussion of climate as a causal factor in Egyptian history, should be edited out or moved to the discussion section. The introduction should also acknowledge previous research on volcanic eruptions, Nile flood levels, and famines in during recent centuries, for which there are Nilometer measurements and abundant historical records—see especially Alan Mikhail, 'Ottoman Iceland: A Climate History,' Environmental History 20 (2015): 262–84. https://doi.org/10.1093/envhis/emv006. This research, particularly for the Ottoman era, already makes a strong case that volcanic eruptions have had major historical impacts on Egyptian society by causing low Nile flow, shortages, epidemics, and political instability (indeed, a stronger case, with richer detail, than is possible for ancient history).**

We have now included reference to Mikhail's work, as well as several other authors studying human-environmental relations in both earlier and later periods of Egyptian history. These include:

Bell, B. "Climate and the History of Egypt: The Middle Kingdom," American Journal of Archaeology 79/3 (1975): 223-269.

Butzer, K. W. "Long-term Nile flood variation and political discontinuities in pharaonic Egypt." In: From Hunters to Farmers: The Causes and Consequences of Food Production in Africa. Eds. Clark, D. and Brandt, S. A. Berkeley, 1984, pp. 102-112.

Hassan, F. "Nile Floods and Political Disorder in Early Egypt." In: Third Millennium BC Climate Change and Old World Collapse. Berlin: Springer, 1997, pp. 1-23.

Hassan, F. "The Dynamics of a Riverine Civilization: A Geoarchaeological Perspective on the Nile Valley, Egypt", World Archaeology 29(1) (1997): 51-74.

Said, R., The River Nile: Geology, Hydrology, Utilization. Oxford, 1993.

McCormick, M. "What climate science, Ausonius, Nile floods, rye, and thatch tell us about the environmental history of the Roman Empire." In: The Ancient Mediterranean Environment between Science and History. Ed. Harris, W. V., Brill, 2013, pp. 61-88.

**1.2. The real question is whether this was also the case in the Ptolemaic period. The article's arguments regarding attribution of societal impacts to volcanic eruptions are often imprecise. I would stress that the attribution of societal impacts to climate variability should be as clear and logical as the attribution of climate impacts to climatic forcings. In this case, the authors aim to evaluate whether and to what extent volcanic eruptions were responsible for revolts in Ptolemaic Egypt. They have made a prima facie case for a causal connection in previous research, which demonstrated a correlation between the timing of eruptions and timing of revolts. Now they are taking this causal argument one step further.**

The reviewer in fact expresses one of the underlying goals of our project very well here: that the attribution of societal impacts from volcanic eruptions in our study period/region should be as clear as the attribution of hydroclimatic variability from these eruptions. We are, however, now clearer in stating that this is not the ultimate goal of the present paper.

Thus, we state in the Introduction: *"In this study, our main intent is to advance our understanding of the likely hydroclimatic impact of his eruption quartet as a foundation for further work aimed at establishing the nature of the causality underlying the observed association between volcanic eruptions and Ptolemaic-era internal revolts."*

Given this, in the present paper, our contextual discussion of the potential role of the hydroclimatic variability (which our modeling results now bring into much greater clarity) must for now be expressed in more contingent and conditional terms. That said, as per our response further below, we have added a more explicit statement on what our work in previous papers has done to date (by way of establishing a causal link between eruption-induced hydroclimatic variability and revolt) and what remains to be done.

**1.3. In this regard, the article should first specify its causal argument(s), preferably in contrastive terms. (For more on this issue, see e.g., S. White and Q. Pei. 'Attribution of Historical Societal Impacts and Adaptations to Climate and Extreme Events: Integrating Quantitative and Qualitative Perspectives'. Past Global Changes Magazine 28, no. 2 (2020): 44–45. https://doi.org/10.22498/pages.28.2.44 ) Do the authors mean to argue that the presence (rather than absence) of volcanic eruptions caused the occurrence (rather than non-occurrence) of revolts? Or do they mean to argue that the timing of the volcanic eruptions explains the timing of the revolts (which may have occurred anyway but in different years)?**

**Or is it some other distinction about the eruptions or climate forcing that explains some other difference in societal impacts? I would stress that these are each very different arguments (though not mutually exclusive). They each require different evidence and each have different implications for Egyptian history. Until the authors specify which causal argument(s) they are making, it is difficult to determine whether they have succeeded or failed.**

We thank the reviewer for their reflection on the nature of possible causal linkages and characteristics. We have now included several citations to White and Pei's (2020) valuable framing paper in attempting to better clarify the contribution of the present paper, and how it may contribute to future research into establishing and characterizing the causal relationships between sudden hydroclimatic variability and various political and socioeconomic behaviors in Ptolemaic Egypt, including revolt. See also our response to the point below.

**1.4. If the article intends to determine whether and to what extent the occurrence of eruptions were responsible for the occurrence of revolts in Egypt, then that will require a more clear and rigorous approach to causation. To clarify this problem, and to avoid some of the confusion that often clouds discussions of climate impacts on human societies, it may help to use a simple analogy. Let us suppose a doctor prescribes vicodin (v) to a bus driver without offering appropriate warnings about its side effects. The bus driver subsequently causes a road accident in which another driver is injured. The injured party sues the doctor on the basis that the negligent prescription (v) caused erratic driving (d) and therefore the accident (a) and the injury (i). In common law, to demonstrate the doctor's responsibility for the injury the injured party would have to demonstrate with a preponderance of evidence at least the following two points: First, that the negligent prescription for vicodin was specifically necessary for the injury to**

occur (i.e., the "but-for" test). Second, that negligently prescribing medication is somewhat sufficient to cause injuries in general (i.e., the "harm within risk" standard). We could also express these two causal chains as two sets of conditional probabilities that would have to meet a reasonable threshold: first, $p(v|d)$, $p(d|a)$, $p(a|i)$ and second, $p(D|V)$, $p(A|D)$, $p(I|A)$, where lowercase letters stand for specific real-world events and the capital letters stand for a type of event in general. These legal standards capture everyday understandings of causation and responsibility as well as centuries of philosophical discussion and legal experience.

While all this might seem a long way from volcanoes and instability in Ptolemaic Egypt, the issue of attribution here is basically the same. To what extent was a volcanic eruption (v) responsible for political instability (i), throughout the mechanisms of drought (d) and famine (a)? To attribute the political instability to the eruption, a preponderance of evidence should demonstrate a strong chain of specific necessity and at least a weak chain of general sufficiency from (v) to (d) to (a) to (i). If there were alternative sufficient causes and the eruption was not necessary for the outcome—let's say another climatic event would have caused a drought even in the absence of an eruption—then we cannot attribute the societal impact to the volcano at all. If the chain of causation depended on extraordinary contributory factors—let's say the Ptolemaic empire was unusually reckless or vulnerable to instability (not wearing its seatbelt, metaphorically speaking)—then the causal responsibility of the eruption would be much diminished, and it would be misleading to refer to the eruption, rather than weaknesses within the empire, as "the cause" or even "a cause" of the occurrence of revolts. Much of the historical discussion in the paper suggests this may have been the case.

We again thank the reviewer for this commentary, and we are particularly happy that it is accessible as a guide to others given the open peer review format of Climate of the Past. In our revisions, we have now placed more explicit emphasis on the importance of establishing and qualifying the character of causality in future work, such as in our planned follow-up case-study paper. For example, in our Introduction, we now state:

*"For Ptolemaic Egypt, the temporal correspondence between internal revolts and explosive volcanism certainly appears recurrent and non-random (Ludlow and Manning, 2016, 2021; Manning et al., 2017; Izdebski et al., 2022). That the revolts and volcanic eruptions under study are known from different archives with independent chronologies (historical documentary and ice-core) has also helped to exclude potential biases in estimating this statistical significance. For example, inflated positive correlations may result when events are known from the same sources (e.g., between extreme weather and societal stresses such as famine or disease, if those instances of extreme weather that contributed to such stresses were more likely to have been documented than those that didn't (White and Pei, 2020)). While the results of Ludlow and Manning (2016, 2021) and Manning et al. (2017) thus imply a causal linkage between explosive eruptions and Ptolemaic-era revolts, much work remains to determine its underlying character, including how direct or indirect it may have been, whether this changed*

*meaningfully between revolts (which varied in date, geography and scale), and (relatedly) what pathways were in effect to "operationalize" any such linkage. Answering such questions is now deemed a key challenge for climate historians and related scholars (White and Pei, 2020). Taken alone, such a correlation does not establish (nor necessarily even imply) causation. Causality is, however, at least implied in cases where analyses are conducted alongside statistical significance testing, with the resulting correlations considered unlikely to have arisen purely by chance, and when such results are interpreted with reference to the relevant historical context, allowing causal "pathways" to be credibly hypothesized (Izdebski et al., 2022).*

**1.5. What this study has done is to take a one small but important step toward demonstrating potential causal responsibility of volcanic eruptions for Egyptian instability by demonstrating the causal sufficiency of eruptions for Nile droughts in general: p(D|V). The paper needs to put this contribution in perspective and not claim to do either more or less. It should neither hide nor exaggerate the significance of this contribution with vague language about volcanoes "playing a role" or an "environmental context" for the disaster.**

As per our response to 1.2, we have been deliberately careful in our use of language precisely because it is beyond the scope of the present paper to ultimately delineate the character of the potential underlying causality which (agreeing with the reviewer) is not likely straightforward. In our revisions, we have now emphasized that the goal in future work will be to move to a greater precision in specifying causality than is currently allowed. We also better emphasize (as stated previously) that the modeling results presented here, by informing us of the likely magnitude and persistence of the hydroclimatic variability experienced in the 160s BCE, will provide an important aid to this effort, and that this is the main intent of the paper.

**1.6. It is entirely possible that we could one day demonstrate that volcanoes were causally responsible for revolts in Egypt, with similar standards and rigor that courts use to assign legal responsibility for damages. This is more than "playing a role": it is causal responsibility. However, this would require further research into other steps in those causal chains, including comparisons with better documented episodes during the medieval and Ottoman eras. On the other hand, if there were alternative sufficient causes of the drought, famine, or instability, or if Ptolemaic Egypt only faced problems because it was extraordinarily vulnerable, then it does not make sense to talk about the eruption as the cause of revolts at all (except perhaps as a trigger for the timing of the revolts). Talk about "a role" for the eruptions would be more misleading than helpful. Nor does it help to include additional historical context (i.e., lines 795-843) if that context is not clearly addressed to a causal argument. If the authors intend to state that there were (or were not) alternative sufficient causes for Egyptian revolts besides eruption-induced droughts, then they should state that clearly. If they intend to state that changes in Egyptian leadership explain why some eruptions were followed by revolts but others were not, then they should also state that clearly. Otherwise, readers are left to infer causal arguments where the authors may not**

**have intended them and where they may not be warranted. I can see that the authors are aiming for greater subtlety and sophistication; however, additional information that is not clearly tied to the causal argument(s) creates more confusion than clarity. Clearly, this study cannot yet provide a definite answer to the question of causal responsibility of volcanic eruptions for the occurrence (or is it timing?) of Egyptian revolts—nor does it need to. However, the authors need to be clear what contributions they can make to this question: that is, how we may update our assessments of the probabilities of necessity and sufficiency along relevant chains of causation. They may also explain what questions remain to be answered and how further research might address them.**

We thank the reviewer for the continued constructive guidance here, which will be put to good use in our planned follow-up paper, which will undertake a case study of the "role" of the volcanic quartet of 168-158 BCE in the revolts and other major societal stresses of this period of Ptolemaic history, building upon the insights provided by the modeling in the present paper. In this follow up, explicit attention will be paid to the causal character of this role.

**1.7. The sections on climate modelling are mostly beyond my area of expertise to evaluate. However, with respect to evaluating historical societal impacts, I would question the emphasis on mean precipitation anomalies. To evaluate whether eruptions were a sufficient cause of a low Nile flow, what I really want to know is how much more probable a low Nile flow would be with an eruption vs. without an eruption: p(D|V)/p(D|¬V). That is, I need some help in assessing the counterfactual scenario: if there hadn't been those eruptions, would there probably have been droughts in Ptolemaic Egypt anyway? The conclusion on lines 578-580 ("likely to have strongly influenced") is too vague. The crucial issue in attributing societal impacts to volcanoes is just how likely it was that deficient Nile flows occurred due to eruptions.**

We agree that this is an important consideration in the assessment of causality. We have emphasized in our revised manuscript that the Nile summer flood was famously mercurial, and that historical explosive volcanism was responsible only for "some" of this variability. We have also cited important precursor work (Manning et al., 2017) using the Islamic Nilometer that has shown tropical and extratropical eruptions to be repeatedly associated with a below-average summer flood, i.e., lower Nile floods were more likely in "volcanic years" than "non-volcanic years".

For the present paper, however, the intent of the modeling is to provide an assessment of what likely happened to the Nile flood given that we do at least know (with fair confidence, thanks to the improved ice-core volcanic forcing history of Sigl et al. (2015)) that four notable eruptions *did* occur.

As part of this, considerable attention has been paid to specifying appropriate conditions for the period in terms of vegetation cover and other forcings that will have mediated the impact of these eruptions. Perhaps more germane to the reviewer's comment here is

that in conducting multiple model runs, we have some additional insight into the range of possible Nile flood responses to this eruption sequence, and have noted occasions when there is high variability among model ensemble members (i.e., notable departures from the mean response).

**1.9. Much of the material currently in the introduction and results sections reads more like discussion. I would encourage the authors to create a larger discussion section in two parts: one for the discussion of volcanic forcing and hydroclimate anomalies and another for discussion of societal impacts. The article would also benefit from a real conclusion that summarizes findings and returns to issues raised in the introduction.**

As noted earlier, we have now revised the paper, including by cutting a substantial portion of introductory historical context, while in the Discussion and Conclusion, we circle back to reflect upon the issues raised in the Introduction, including with a more explicit statement on the need to go further in assessing historical causality.

**1.10. The authors may also wish to address the methodological significance of the work and, in particular, make proposals for further integration of paleoclimatology, climate modeling, and human history.**

We agree fully that an increased integration between palaeoclimatology, climate modeling and human history is an important methodological goal. We have taken the opportunity provided by this paper to note that climate modeling can make a tremendous contribution to our understanding of human history, in particular by providing insight into the mechanisms by which events like distal explosive eruptions might impact agriculturally critical environmental resources like the Nile summer flood, and by filling in the "blanks" for periods and regions when and where palaeoclimatic proxies (natural archives) are not available in abundance or at sufficiently high temporal and spatial resolutions. We also note the importance of developments in palaeoclimatology for the study of environmental influences on society, in particular the important work of the PAGES 2k Consortium. We then note that extending reconstructions beyond the nominal "2k" target period would help provide environmental data for some of the most well-documented societies of ancient world.

**1.11. Specific issues: Line 15: The phrase "sometimes widespread" is confusing. Based on context, I would suggest "both local protests and widespread revolts".**

We have kept the use of this phrase, placed in parentheses, because the events in question appear to have been more substantial than local protests, sometimes taking the form of organized attempts at the overthrow of Ptolemaic rule. To help give the reader a greater grasp of the potential scale of these events, in our Introduction we cite the example of the Great Theban Revolt that lasted approximately twenty years and in which the Ptolemies appear to have lost control of much of southern Egypt.

**1.12. Line 24: I assume that "observe" here refers to finding an average in the simulations, not an actual observation of the real climate. Please clarify.**

By "observe", we mean here that we are observing (reporting) that our model produced an average (mean) surface cooling of the order of 1.5C following the first (tropical) eruption in 168 BCE. We have kept this term, but have made multiple textual edits to the manuscript for purposes of clarity (detailed in the Track Changed manuscript).

**1.13. Line 55: This statement already presupposes the conclusion.**
We feel that stating that Egyptian civilization was heavily dependent on the Nile is relatively non-contentious, and we mean this in a general sense more broadly for Egyptian history than solely for the Ptolemaic period that we are studying.

We have added multiple additional citations (see earlier) that have studied the inter-relations between Nile flooding and Egyptian civilization in different periods, including the reviewer's valuable recommendation of Mikhail, A. 2015. 'Ottoman Iceland: A climate history', Environmental History 20: 262–284.

**1.14. Line 56: The phrase "potentially climatically effective" is awkward. I would recommend perhaps "eruptions that may have had regional or global climatic impacts."**

We thank the reviewer for their recommendation. Respectfully, we have maintained the use of this phrase as being slightly more concise. The phrase "climatically effective" is also relatively common in the volcano-climate literature to denote those minority of eruptions having the characteristics capable of impacting climate on more than local scales.

**1.15. Line 57-58: Again, this statement presupposes the conclusion.**

Rather than stating that *"Egyptian civilization may have been repeatedly influenced by the "hydroclimatic shocks" wrought by these events (Manning et al. 2017)"*, in our revisions, we now state more carefully that *"Egyptian civilization provides a valuable test-case for the study of human vulnerability and resilience to abrupt environmental change in potentially experiencing repeated "hydroclimatic shocks" induced by these events (e.g., Manning et al., 2017, 2021)."*

**1.16. Line 146-152: I do not find that this example supports the authors' arguments. Instead, it serves as a reminder that there were, at times, other sufficient causes of political change in Egypt besides climatic variability, such as conflicts with neighbouring empires.**

This portion of text has now been cut from the Introduction as part of our efforts to condense the overall size of that section.

---

## Author Comment (AC2)

We thank the reviewer for their valuable comments. We present below the comments in black, our replies in blue, and the changes made to the text of the manuscript in blue italic font. Line numbers correspond to the original manuscript as well as the new revised manuscript.

General: The manuscript investigates the hydrological response of a series of volcanic eruptions during the 2nd century BC, focusing also on the societal impacts of the eruptions in the context on the Egyptian history. In addition to empirical evidence, authors use the output of an ensemble of simulations with a comprehensive Earth System Model, simulating potential trajectories of plausible climatic scenarios in the aftermath of the volcanic eruptions.

The manuscript is very well written, material and methods are comprehensively presented and results are discussed within the context of present literature. Therefore, I think the manuscript should be published with some minor comments addressed below. Most specifically, the comments relate to the modeling and statistical part, including a more nuanced discussion and interpretation of model results in the context of past civilizations. Moreover, I would encourage the authors to reduce the overall length of the manuscript by summarizing dedicated paragraphs or moving parts into the supplementary material whenever possible.

The manuscript is revised as described in detail under the specific replies below. We have reduced the main manuscript length as recommended. Some of the old text has been deleted, while other text has been moved to the supplementary material. We note that the manuscript is still comparatively extensive, but this is in line with other published article in this special issue, which (given their interdisciplinarity) have had to introduce a wider range of methods and contexts than might otherwise be required.

1 Introduction:
l. 114: Linking volcanic eruptions directly to revolts or warfare might be afflicted with high degree of uncertainty: In past societies single upheavals or riots always happened – likewise, a close inspection of ice core records will typically also yield one or two eruptions per decade. Linking both just because of their synchronicity might be co-incidental. The processes of both, the impact of the volcanic eruption on climate and the prerequisites leading to riots or revolts during the period previous to the volcanic eruption can have multiple drivers and causes. Therefore this line of evidence in terms of wiggle matching single historical events with volcanic outbreaks should be handled with care.

We fully agree with the concerns listed by the reviewer. In the manuscript, we have now clarified that our work takes place in the context of the previous results of Ludlow and Manning (2016, 2021) and Manning et al. (2017), which considered in detail the correspondence between

explosive eruptions (using the volcanic chronology of Sigl et al. (2015) and independently dated revolts across the Ptolemaic era. This work has identified the correspondence as highly statistically significant, also identifying significant correspondences between dates of phenomena such as sales of hereditary agricultural land, which have been previously hypothesized to occur with greater frequency during times of socioeconomic stress, as might follow years of poor Nile flooding, providing a glimpse into the mechanisms by which the impacts of explosive volcanism on Nile flooding might ultimately provoke revolt.

Scholars have now also recommended that case studies of more specific periods and regions might shed greater insight into the drivers and prerequisites of complex phenomena such as revolts. Our present paper thus intends to provide a foundation for such a case study by offering greater detail about the likely hydroclimatic impacts of a key decade in Ptolemaic Egyptian history, the 160s BCE.

l. 118: "hydroclimatic shocks" should be replaced by "pronounced changes in hydrology"

We have corrected the relevant sentence at line number 118 to state "*pronounced changes in hydrological cycle*". (Line 152 in Revised Manuscript)

l. 120: The hydrological cycle after very large explosive tropical eruptions is not only driven by the north-south contrast of the monsoon (the African monsoon system is far more complex in this respect), rather than less evaporation caused by lower temperatures according to the Clausius-Clapeyron equation.

We have modified the relevant sentence in line 120, to acknowledge the role of regional factors in addition to the reduced temperature gradient *"....well as reducing the meridional (north-south) temperature contrast that controls the intensity of the African monsoon, alongside other regional factors*". (Line no. 154-155 in Revised Manuscript)

ll. 134–170: This whole section should be shortened/summarized and focus on the very area of research, as outlined in the section ll. 171–186.

We have moved the a substantial portion of the introductory historical context (lines 134-170) to the supplementary material (now appearing there as Section S1.1 Introduction (Historical Context)). (Reference to supplementary info at line number 175 in the Revised manuscript)

ll. 190–205: This section should also be shortened to the most relevant information introducing the content of the subsections.

We have shortened the section (by deleting some information related to model that is available elsewhere (as per the cited paper) as suggested. (Around the line 200 in Revised Manuscript)

In addition to the points mentioned above there are two additional points that should be mentioned already in the introduction:

1) The importance of natural climatic/hydrological variability in the occurrence of Nile floods and their counterparts. This is important to put the proposed "hydrological shocks" in the aftermath of volcanic eruptions into context of externally undisturbed periods.

This point is addressed in the introduction in lines 90-105. In addition, we have inserted some text and relevant citations to emphasize that indeed the considerable variability of the Nile was well known historically, and that explosive volcanism contributed to "some" of this variability (i.e., it certainly did not drive all the observed variability).

See lines 109 in Revised Manuscript: *"But the Nile summer flood was also famously mercurial, with insufficient flooding often leading to adverse societal impacts (e.g., Bell, 1975; Butzer, 1976, 1984; Said, 1993; Hassan, 1997a, b; Hassan, 2007; McCormick, 2013). Some of this variability was likely driven by explosive volcanism."*

2) A more differentiated introduction of the impact of large explosive tropical volcanic eruptions vs. medium-to-small sized high latitude northern/southern hemisphere eruptions. This relates for instance to the overall amount of cooling, the potential for a dynamical response on natural modes of climate variability (cf. North Atlantic Oscillation/El Nino). Introducing this difference in location and magnitude will also help to better explain the different climatic and hydrological response of the initial tropical eruption E1 and the following high-latitude eruptions E2 – E4 that are presented and discussed further down in the manuscript.

Point 1 and 2 are addressed in the Introduction section where volcanic eruptions and their impacts on the Earth's climate system are summarized (lines 55-75).

In our opening paragraph (lines 52-58 In Revised Manuscript), we thus now state: *"The cooling caused by such events can also reduce net evaporation and hence precipitation over large areas (Lui et al., 2016; Iles et al., 2013), while also potentially leading to a near global-scale dynamical suppression of the northward migration of the inter-tropical convergence zone (ITCZ) during the boreal summer, as the convergence follows the surface area of maximum temperature (Petterson et al., 2000; Chiang and Bitz, 2005; Broccoli et al. 2006; Colose et al. 2016). These changes in precipitation can, moreover, impact river outflow (Oman et al., 2006; Sabzevari et al., 2015; Kostiç et al., 2016)..."*

Additionally, we have added the following in lines 77-84 (Revised Manuscript):

*"Extratropical eruptions generally have a comparatively weaker climate impact than tropical eruptions. This happens* following the background Brewer-Dobson circulation upwelling in the tropics and downwelling at higher latitudes, which *directly affects the stratospheric lifetime of*

*volcanic aerosols (Kirtman et al., 2013; Myhre et al., 2013; Schneider et al., 2009). Recent studies though illustrated the potential for extratropical eruptions having disproportionally strong forcing and climate impacts, consistent with past reconstructions (Toohey et al. 2019).*"

2.1 Model Description

l. 226: How is the impact of volcanic eruptions implemented?

We have provided a paragraph on how the eruptions were implemented in the NASA GISS ModelE was originally stated under the section "Experiment Design" (line 272 in original Manuscript), We moved this paragraph from experiment design to discuss how the volcanic eruption has been implemented here (Line 239-252 in Revised Manuscript).

2.2 Experiment Design

l. 233: What is the rationale [for] using the PMIP4 mid-Holocene protocol? Maybe the authors could explore in one or two sentences why especially the vegetation is closer to mid-Holocene conditions rather than the one representing the situation during pre-industrial times.

NASA GISS ModelE's Terrestrial Biosphere Model (TBM) Ent (NASA-GISS Version name) (Kiang, 2012; Kim et al., 2015) is not a full Demographic Global Vegetation Model (DGVM). A key missing functionality is the ability to migrate vegetation, driven by changes in climate. In CMIP5, the lack of a fully dynamic vegetation model GCMs led to a failure to reproduce the mid-Holocene wet Sahara conditions over Africa (Harrison et al. 2013). Our model simulations for the mid-Holocene period using the vegetation distribution based on the PMIP4 protocol produced a more realistic result in terms of a wetter Sahara region. Thus, in the absence of a better approach, we linearly interpolated the mid-Holocene PMIP4 vegetation distribution to the 2.5k period to achieve more accurate background climate conditions.
We have outlined our arguments for the use of the PMIP sensitivity vegetation distribution in our results section while discussing the implications of using these boundary conditions in section 3.1.1, ( line 300 in Revised Manuscript and specific discussion is at line 340 onwards).

l. 272: The authors could add some effects on the timing of the eruption, i.e. when the eruption date is set to a summer date, especially for the potential effects on monsoon and the northern hemispheric winter atmospheric circulation. It could also be explicitly stated that it is not possible to decipher the exact timing of the eruption in the annual cycle because of dating uncertainties involved in the ice core reconstructions.

Thank you for pointing this out; we have thus added the line "*Because the exact date of an eruption cannot be directly determined based upon ice-core sulphate deposition data, both because of possible uncertainties in the ice-core chronologies and because of variable time lags between eruptions and the atmospheric circulation of the resulting sulphate and its deposition in the polar ice, we selected a summer eruption date to investigate the impact on northern hemisphere monsoon and wintertime atmospheric circulation.*" (Line 291 in Revised Manuscript)

l. 273: I suggest to move the following section on the implementation of the volcanic forcing at the end of the model description paragraph – also some words on the uncertainties of the sulfate reconstructions based on ice cores would be helpful to indicate that modeling results on the subsequent simulations are dependent on the magnitude of the reconstructed sulfate injected into the stratosphere.

We  moved the relevant paragraph to the model description section as suggested, and added following lines at line 295.

"*We also note that the accuracy of our modelling will depend in part upon the accuracy of the ice-core-based volcanic forcing reconstruction being employed. Uncertainties in reconstructed forcing can arise, for example, because of variation in the deposition of sulphate across the polar regions for any given eruption. In this respect, it is important to note that the Sigl et al. (2015) volcanic forcing reconstruction employs several ice-cores from Antarctica and Greenland, but our results can be revisited as reconstructions become more reliable by incorporating larger numbers of ice-cores*" (line 295 in Revised Manuscript)

3. Results

– The header for paragraph 3.1 is missing –

We have introduced the header for 3.1 as "**3.1 2.5ka control runs**".

Changes in orbital forcing – the supposedly most important factor between 2.5 and 6k – were already considerable different at 2.5 k. Therefore I guess that also the classical mid-Holocene pattern is different, even without dynamic vegetation. It would be good to at least indicate those implications when interpreting the 2.5 k pattern in the context of the mid-Holocene 6k climate and vegetation changes.

Another note: Changes due to orbital forcing are mostly effective on a seasonal basis on Holocene timescales, because changes in the inclination of the earth axes do not change the annual amount of radiation received by the sun. An alternative in structuring Fig 1 and Fig 2 is to omit the mean climate states in the upper and middle panel (also for section 3.1.2) and replace them by the patterns for the winter and summer season for the different experiments (together with the annual). This would also show better the impact of the (orbitally induced) background climate conditions between 2.5 k and PI.

We certainly agree that the impact of orbital forcing for mid-Holocene may be slightly different (cooler for mid-Holocene) than the PI control for both the current (PMIP4-CMIP6) and previous (PMIP3-CMIP5) generation of models (Brierley et al., 2020). We have focused on the North African monsoon season rainfall and the impacts due to the inclusion of PMIP4 vegetations over

the region. Since the impact of changes in orbital forcing for a 2.5k period is evident in the surface temperature changes over the northern hemisphere, but rainfall changes over Africa appear more reasonable with vegetation changes only.

We have now modified the plots (Fig 1 & Fig2). We have included the seasonal differences for Annual, DJF, and JJAS seasons along with the mean seasonal climate for the 2.5ka period with GHG and ORB in fig 1 and GHG, ORB, and vegetations in fig 2. We included the mean panels in both plots for the reader to understand the mean climate with the difference with the inclusion of different forcing factors. See the revised Fig 1, below.

[Figure]

"Fig 1. *Seasonal means (Annual, DJF & JJAS) of surface air temperature (top row) for 2.5k period equilibrium run, differences from the preindustrial period (2.5ka-preindustrial) for all three seasons (2nd row from top) and seasonal (Annual, DJF & JJAS) mean precipitation (3rd row from top) and the difference (bottom row) from preindustrial period (2.5ka-preindustrial). The equilibrium run for 2.5k period have the orbital and GHG concentration changes for the 2.5k period (referred as OG), the preindustrial period (as PI), and their difference (OG-PI) as simulated by GISS ModelE2.1.*"

For the revised Fig 2, see below:

[Figure]

*"Fig 2. Mean surface air temperature for Annual, DJF and JJAS seasons (top row) and seasonal mean precipitation (3rd row from top) for the equilibrium runs with the PMIP4 vegetation for 2.5k period and surface temperature difference (2nd row from top) as well as the seasonal precipitation differences (bottom row) for 2.5k period as simulated by GISS ModelE2.1. We used a short initial notation for forcing to denote the difference (ORB+GHG+VEG = OGV and ORB+GHG= OG)"*

3.1.2 2.5Ka ORB+GHG+VEG climate

l. 344: The authors should provide some implications the linear interpolation of vegetation might have on their results (e.g. it is also likely that vegetation changed considerably earlier to preindustrial-like conditions, resulting in a higher albedo due to less forest over the high northern latitudes.)

We have thus summarized the implication of the inclusion of vegetation cover specific to 2.5ka as increased temperature over higher latitudes and northward expansion of the African monsoon during the JJAS season. Although the albedo changes due to introducing vegetation over Africa are not substantial, this enhanced rainfall supports the role of biogeophysical processes in reproducing the wet African conditions over Africa relative to PI period for the mid-Holocene

period (Kutzbach et al., 1996; Claussen et al., 2003; Kutzbach and Liu., 1997; Hewitt and Mitchell, 1998). These results also support the importance of having a dynamic vegetation component to represent regional-scale processes and their impact on the climate.

3.3 Volcanic aerosol properties

Concerning the overall length of the manuscript, I suggest to move this section into the supplementary information, as the general content of the manuscript is for an interdisciplinary readership.

We prefer to keep this section in the manuscript because it conveys important information on the model setup, which might get lost in the supplement. This section also complements the description section on how we model sulfate aerosols, aerosol-radiation interaction, and related properties. We also note that the journal does not have strict word limits, and our article is not overly long relative to others in the same special issue. We of course take the reviewer's general point that the manuscript should not be needlessly long.

3.4. Latitudinal temperature response to volcanic aerosol forcing

l. 483: How did the authors estimate their level of significance ? A few words in the supplementary [material] or within the section would be helpful to assess the robustness of the test, using only a limited number of ensemble simulations for the estimation of the level of statistical significance.

Thanks for pointing this out. To highlight this, we have added additional sentences in section 3.4 along with our results:

*"The statistical significance level is estimated using the 2-tail student t-test after Deser et al., (2012) and following the assertion that 10 ensembles are sufficient for reasonable estimation of internal variability at a regional scale (Singh and AchutaRao, 2019)."* (Line 484 in Revised Manuscript)

3.5 Latitudinal precipitation response to volcanic aerosols

ll. 506: The authors should add one or two sentences on the potential complications [of] investigating the direct output of global and coarsely resolved earth system models. For instance, the simplified parameterizations used for the simulation of precipitation in global models which impact a realistic simulation of tropical convection.

The coarser resolution of Earth system models is a notable cause of uncertainty in the modeling of convective rainfall. However, recent finer resolution models with convective cloud resolving capabilities have shown a significant improvement relative to coarse resolution models. But coarser resolution models can still be successful in simulating large-scale patterns of changes in rainfall. We have thus added these lines:

*"We used a coarser resolution earth system model having a simplified parameterization and was successful in simulating the large-scale patterns of rainfall change"*.

l. 514: This section is one of my critique zones, especially in the context of interpreting climatic trajectories in the context of past societies: The ensemble mean never happens in the real world – if any, a single trajectory compares best to a real world manifestation. Therefore it would also be imperative to show trajectories for single ensembles. This also reflects the bandwidth of potential hydrological changes in the aftermath of volcanic eruptions.

The reviewer's point. The ensemble mean may, for example, be biased by several factors, such as outlier members of the ensemble. One of the reasons for focusing on the ensemble mean is that it is impossible to select the most accurate member as we do not have the historical observations to compare to. However, the spatial representation of ensemble means with accompanying statistical confidence information can be helpful in the interpretation of the robustness of the magnitude and sign of change across ensemble members.

To convey how the results of individual members may differ meaningfully from the ensemble mean with, for example, stronger variability in the signal for a field such as rainfall, we have now plotted two (see below) ensemble members and compared them with the ensemble mean. These two individual members do exhibit some difference in rainfall response in the northern hemisphere tropics after the eruption E1. Still, we feel that in the absence of any indication of which ensemble member is the more accurate, the ensemble mean is the most relevant to focus on in the main text, but we explicitly note that it may be the case that one or more of the ensemble members is more accurate, though we cannot at present tell which. We have added the following text to the manuscript:

*"It can be argued that an individual ensemble member can represent the historical period, but it is impossible to select in the absence of observation. Also, the added noise due to natural variability can alter the sign of change at the spatial scale among the individual ensembles. Thus, we selected the ensemble mean with statistical significance to show the response to volcanic eruptions with robustness for the climate variable. "*

To additional reflect this uncertainty, we emphasize (see, for example, lines 542-547) cases in which ensemble variability around the ensemble mean is particularly high, thereby alerting the reader to instances in which the ensemble mean may be less parsimonious.

[Figure]

l. 540: Here again, a more detailed information on the evaluation of the statistical significance would be helpful.

Please refer to our reply to the reviewer's comment on section 3.4, above.

3.6 African monsoon and Nile River response

l. 581: The already mentioned information that a more consistent comparison between model and empirical evidence can only happen at the single simulation level can again [be] picked up here, because in the real world one could not expect a mean response of different simulated trajectories for single events in history.

Please refer to our response to the reviewer's comment on section 3.5, above.

l. 584: Results for the E1 eruption seem convincing and also have a large-scale character that can be attributed to an external event – However, eruptions E2 – E4 show a very inconsistent pattern (even in the ensemble mean). This is also reflected in the statistical test (that presumably uses standard testing techniques that are not taking account the small sample size of n=10 samples). This heterogeneity in the response of the northern hemisphere E2 – E4 eruptions should be more emphasized, also in the subsequent interpretation in the context of their sustained effects on Nile floods.

This is an important observation. We have now inserted two new paragraphs along with the discussion on the possible reasons for heterogeneity in the response over the Nile basin.

(Line 743-785 In the Revised Manuscript)

"*It is evident that the mean surface temperature response in the northern hemisphere is significant at the control period's $1\sigma_{ctrl}$ and $2\sigma_{ctrl}$ levels. However, while rainfall and river discharge responses are significant at the $1\sigma_{ctrl}$ level, they fall within the $2\sigma_{ctrl}$ levels, although a few individual members do show significance at $2\sigma_{ctrl}$ as well. However, the statistical significance of the rainfall and discharge response may be sensitive to the dearth in the modeling Nile River basin at a relatively coarse resolution of the GISS ModelE, as well as the boundaries chosen to model the Nile basin and its headwaters. In particular, given the complexity of the Nile's hydrology and disparate sources of discharge for the White and Blue Niles. We thus investigated the post-volcanic change in river flow for the southern (White Nile-dominated) and northern (Blue Nile and Atbara river-dominated) parts of the basin by dividing it at 10° N (Fig 13). Annual mean river flow change for the south (blue lines) and north (red lines) of the Nile basin were in broad agreement with a negative flow anomaly after eruption E1. This was most notable in the eruption year and the first year following, with the 95th percentile envelopes (dotted lines) deemed significant at the 95% confidence level for both these years (i.e., crossing*

*the dashed lines parallel to the x-axis (Fig 13). In contrast, the mean north and south responses disagreed, including in the sign of the observed changes, after the extratropical eruptions (E2, E3 & E4). More specifically, while the mean flow anomalies in the year of E2 were unremarkable and showed little north-south contrast, a more notable divergence was observed in the first year following, with a positive flow anomaly in the south and negative in the north. In the year of E3, flow in the south showed no notable anomaly, while flow in the north was marginally negative. This distinction became more marked in the first year following, mainly due to a larger negative anomaly in the north. In the year of E4, a negative anomaly was again observed in the north, persisting for three post-eruption years, and contrasting with positive or unremarkable anomalies in the south.*

*These results are consistent with our earlier-described results (e.g., spatial rainfall variability over the Nile River basin, as per Figs. 10 and 11) and proposed mechanisms, alongside expectations from the literature (e.g., Manning et al., 2017). Thus, tropical eruptions (like E1) may result in a more consistent (negative) north-south flow response due to their more even interhemispheric aerosol burden and associated radiative impact. Extratropical NH eruptions (like E2-E4) that can result in a more asymmetric hemispheric aerosol burden may, by contrast, introduce contrasting flow anomalies by suppressing the northward migration of the ITCZ, negatively impacting flow in the Blue Nile and Atbara rivers by diminishing monsoon rainfall in the Ethiopian highlands, while potentially enhancing flow in the White Nile, fed by rainfall over the equatorial lakes"*

[Figure]

River Flow Changes for Northern and Southern part of Nile Basin
(Season:ANN)

*Fig 13. Annual Nile River flow changes averaged over the northern (red) and southern (blue) parts of the basin (divided at 10° N) for the entire simulation period. The solid lines represent the ensemble mean for each part of the basin; the dotted lines are ±1.95σ, where σ is derived from across all the ensembles, and the horizontal dashed lines parallel to x-axis are the ±1.95σ$_{ctrl}$ where σ$_{ctrl}$ is the standard deviation across the 100-year control run. Red and blue lines correspond to the northern and southern parts of the Nile basin, respectively.*

Please refers to the explanation for next comment (l. 614 incl. Table 2: Interpreting….)

l. 614 incl. Table 2: Interpreting the Table and the calculation of the according values correctly, the standard deviation is based on the volcanically forced ensemble members and the difference on the mean over all ensemble members minus the climatological mean of the 100 year control?

An alternative is to calculate the annual standard deviation of the 100 year control run and include it as the 1.95*std = 95% confidence interval. This will give an indication how the mean discharge value is outside the natural range. In the present form it gives the bandwidth of the volcanically forced simulations, not taking into account the natural undisturbed variability. The interpretation of the 95% confidence interval based on the control will give an indication how exceptional the respective year after eruption E1 – E4 was in the context of the natural variability.

We thank the reviewer for this suggestion. We have calculated the variability (σ$_{ctrl}$) for the 100-year control period and compared the statistics in Table 2 against the confidence interval (1.95*σ$_{ctrl}$) suggested by the Reviewer. The calculated variability (σ$_{ctrl}$) and 95% confidence interval (1.95*σ$_{ctrl}$) for annual river flow are 25.20% and 49.155% respectively. It is noticed that annual ensemble mean change is within the 95% confidence interval, but individual ensemble member reaches beyond 95% confidence interval for some years. River flow change over the Nile basin varies up to 3*σ for a few ensembles for some years.

[Figure]

Fig Rev2.1. The solid blue line shows the annual mean change in river discharge over the Nile River basin, and dashed (blue) are the individual ensemble member. The red-colored dashed line parallel to the x-axis represents the 1.95*σctrl for the 100-year control run.

Fig 12: This figure contains basically very good information – Similar to Table 2, and to show the exceptional behavior of the different metrics, it would be better to illustrate the 1.95*std of the natural 100 y control run as two lines parallel to the x-axis, together with the individual trajectories of the 10 volcanically forced simulations. When a considerable number of trajectories falls above or below the 95% confidence levels for an individual year, one can speak of a robust response – according to the hypothesis proposed, the discharge trajectories should then fall below the lower boundary for the years after the volcanic eruptions.

In addition, without the green vertical lines it is difficult to decipher the volcanic eruptions based on the evolution of precipitation and discharge, because also other sub-periods with considerable reductions in stream flow appear that are unaffected by volcanic forcing (e.g. year 159). An alternative interpretation that could be hypothesized relates to an increased intra-ensemble variability after volcanic eruptions compared to undisturbed conditions.

We have now inserted the 1σ (Red dashed line) and 2σ lines (Red solid line) on the plot for all three diagnostics. The modified Fig 12 is now shown below.

[Figure]

*Fig 12: Monthly time series of individual ensemble and mean of surface temperature response (˚C) averaged over northern hemisphere (NH) (top panel), rainfall change (mm/day) over the spatial box over Nile River watershed (Latitude: (5N, 18N), Longitude: (30E, 42E)) (middle panel) and Nile River discharge anomaly (%) at the delta region (grid box centered at 29.0N, 31.25E). The dark solid (Thick) line shows the multi-ensemble mean, individual member (thin line), and the color envelope shows the associated variability (±σ; Standard deviation). The annual cycle of climate variability of control run is shown as $1\sigma_{ctrl}$ (Red dashed line) and $2\sigma_{ctrl}$ lines (Red solid line) along the x-axis for all three variables. The vertical dotted green line shows when each eruption happens*

4. Discussion and Conclusions

ll. 712: This paragraph also relates to the interpretation of empirical evidence in the context of earth system and climate model simulation: It is important also taking into account the natural or stochastic nature of historical processes that are not always determined by external environmental forcings. Otherwise a state in the interpretation and explanation of historical events will be reached, where numerous single historical events are only interpreted within the climate-determinism concept, which can be true

for severe events, but might be misleading for most medium-to-small size events, especially in the context of volcanic eruptions.

We have now added a more substantial introduction to the Introduction section to address challenges in assessing connections and causality between environmental forcings and historical human/societal events. This also in part addresses similar challenges highlighted by Reviewer 1, and which are addressed more fully in our response to this reviewer.

Additionally, we have clarified that the present paper builds upon the work of Ludlow and Manning (2016, 2021) and Manning et al. (2017), that explicitly address the challenges of testing statistically the association between explosive eruptions and revolts in Ptolemaic Egypt (also noting that it is not assumed that all revolts were "triggered" or otherwise caused by volcanically induced hydroclimatic variability (or indeed hydroclimatic variability of any origin).

The intent of the present paper is, therefore, to provide clarification on the likely hydroclimatic variability experienced in Ptolemaic Egypt during the 160s BCE, a time already well known to historians as one of considerable societal upheaval in Egypt. Our work here should provide a firmer foundation for a planned follow-up study that will more closely examine the available historical evidence for the impacts of these eruptions (now informed by our modelling) and assess their contribution to the turbulent history of the period.

l. 731: For producing a basis for "historical realization" it is again of utmost important to look and investigate the trajectories of individual realizations of ensemble simulations from climate models and not (only) their mean response.

We agree with the reviewer's argument, and would refer back to our earlier responses to this important issue.

l. 791: As the authors state correctly, from a conceptual point of view there is no "best" member, because all members are equal probable under the same set of external forcings implemented. What might be more important is the notion that the combination of external AND internal forcings shape the exact evolution in both, the real and the model world.

We again agree fully with the reviewer here.

Figures and Tables:

In general, Figures and Tables are presented with high quality and an appropriate level of information included. Below just a few minor suggestions how to improve or modify selected items:

Fig 1: As already stated in the main text, Fig 1 and 2 might be combined into one single Figure by representing only the differences for annual, winter and summer (alternative JJAS) mean.

Figures 1 and 2 are now modified to include more seasonal details as suggested.

Fig 5 center panel: The style of the presentation of the single trajectories could be used as template for the precipitation and discharge Figure 12 to show the variability of the different ensemble simulations together with the 95% confidence level of natural variability of the 100 yr control simulation.

Figure 12 has now been modified to include the individual ensemble members along with the $\pm\sigma$ envelope for the volcanically forced ensembles. We have also included the $\pm1*\sigma_{ctrl}$ and $\pm2*\sigma_{ctrl}$ annual cycle for the 100-year control run as suggested.

Table TS1: The authors might include also the volcanically forced simulations as an additional column and highlight those simulations that are presented in the manuscript.

We have inserted the column for the volcanically forced ensemble.

**Additional Revisions**

Apart from the reviewer's comments, we also revised the manuscript sections for some general aspects for better representation of results in the final version of the manuscript and included more relevant references—some of these as mentioned below.

1.) The word "stratovolcanic" has been changed to "Large, strato-volcanic" for more precise reference in abstract line 20 and introduction and 46
2.)  in the abstract "NASA GISS ModelE" is changed to "NASA GISS ModelE2.1".
3.) Abstract line 29: "South and East Asian" changed to "South Asian, and East Asian".
4.) Inserted a sentence at line 48 (introduction) "*The sulfate aerosols of the 1991 eruption of ~18 Tg SO$_2$ from Mt. Pinatubo increased the optical depth of the atmosphere from ~0.6 to ~0.75.*"
5.) Inserted reference "Colose et al. 2016" at line 53.

6.) Text inserted at line number 85. "*Effectively, the ITCZ shifts away from the hemisphere with the greatest amount of volcanic aerosol; for tropical eruptions, this movement is typical more southward as well owing to the larger amount of land in the Northern Hemisphere and higher thermal capacity of the oceans*."

7.) Modified the format of conditions stated at line 250-255.

**Reference:**

Kirtman, B., Power, S. B., Adedoyin, A. J., Boer, G. J., Bojariu, R., Camilloni, I., Doblas-Reyes, F., Fiore, A. M., Kimoto, M., Meehl, G., Prather, M., Sarr, A., Schar, C., Sutton, R., van Oldenborgh, G. J., Vecchi, G., and Wang, H.-J.: Chapter 11 - Near-term climate change: Projections and predictability, edited by: IPCC, Cambridge University Press, Cambridge, 2013.

Ludlow, F. & Manning, J. G. in Climate Change and Ancient Societies in Europe and the Near East: Diversity in Collapse and Resilience (eds Erdkamp, P., Manning, J. G. and Verboven K.) 301-320 (Palgrave Macmillan, 2021).

Ludlow, F. & Manning, J. G. in Revolt and resistance in the Ancient Classical World and the Near East: The crucible of empire (eds Collins, J. J. & Manning, J. G.) 154–171 (Brill, 2016)

Manning, J. G., Ludlow, F., Stine, A. R., Boos, W. R., Sigl, M., and Marlon, J. R.: Volcanic suppression of Nile summer flooding triggers revolt and constrains interstate conflict in ancient Egypt, Nat Commun, 8, 900, https://doi.org/10.1038/s41467-017-00957-y, 2017.

McConnell, J. R., Sigl, M., Plunkett, G., Burke, A., Kim, W. M., Raible, C. C., Wilson, A. I., Manning, J. G., Ludlow, F., Chellman, N. J., Innes, H. M., Yang, Z., Larsen, J. F., Schaefer, J. R., Kipfstuhl, S., Mojtabavi, S., Wilhelms, F., Opel, T., Meyer, H., and Steffensen, J. P.: Extreme climate after massive eruption of Alaska's Okmok volcano in 43 BCE and effects on the late Roman Republic and Ptolemaic Kingdom, Proc Natl Acad Sci USA, 117, 15443–15449, https://doi.org/10.1073/pnas.2002722117, 2020.

Myhre, G., D. Shindell, F.-M. Bréon, W. Collins, J. Fuglestvedt, J. Huang, D. Koch, J.-F. Lamarque, D. Lee, B. Mendoza, T. Nakajima, A. Robock, G. Stephens, T. Takemura, and H. Zhang,: Anthropogenic and natural radiative forcing. In Climate Change 2013: The Physical

Science Basis. Contribution of Working Group I to the Fifth Assessment Report of the Intergovernmental Panel on Climate Change. T.F. Stocker, D. Qin, G.-K. Plattner, M. Tignor, S.K. Allen, J. Doschung, A. Nauels, Y. Xia, V. Bex, and P.M. Midgley, Eds. Cambridge University Press, pp. 659-740, doi:10.1017/CBO9781107415324.018, 2013

Schneider, D. P., Ammann, C. M., Otto-Bliesner, B. L., and Kaufman, D. S.: Climate response to large, high-latitude and low-latitude volcanic eruptions in the Community Climate System Model, 114, https://doi.org/10.1029/2008JD011222, 2009.

Sigl, M., Winstrup, M., McConnell, J. R., Welten, K. C., Plunkett, G., Ludlow, F., Büntgen, U., Caffee, M., Chellman, N., Dahl-Jensen, D., Fischer, H., Kipfstuhl, S., Kostick, C., Maselli, O. J., Mekhaldi, F., Mulvaney, R., Muscheler, R., Pasteris, D. R., Pilcher, J. R., Salzer, M., Schüpbach, S., Steffensen, J. P., Vinther, B. M., and Woodruff, T. E.: Timing and climate forcing of volcanic eruptions for the past 2,500 years, Nature, 523, 543–549, https://doi.org/10.1038/nature14565, 2015.

Toohey, M., Krüger, K., Schmidt, H., Timmreck, C., Sigl, M., Stoffel, M., and Wilson, R.: Disproportionately strong climate forcing from extratropical explosive volcanic eruptions, 12, 100–107, https://doi.org/10.1038/s41561-018-0286-2, 2019.

---

## Author Response (AR2)

Response to comments:

**Investigating hydroclimatic impacts of the 168-158 BCE volcanic quartet and their relevance to the Nile River basin and Egyptian history**

Singh et al. 2022.

We are grateful to the reviewers for their comments and suggestions for improving the manuscript. We believe these have helped us further improve the manuscript. We present below the respective reviewer comments in bold, our responses in red, and use italic font when quoting text added to the revised version of the manuscript.

**Reviewer 2: -**

**Comment** 1.0. **The manuscript in its first version was already in a quite mature state – the revised version includes all the comments raised in the first review in a very comprehensive way. A big asset is the re-structuring of different parts of the manuscript including some modifications of the presentation and interpretation of the statistics related to the ensemble simulation approach used in the study. Therefore I congratulate the authors for the manuscript and suggest to publish it after addressing some minor comments listed below.**

Response 1.0. We thank the reviewer for this assessment.

**C1.1. The first comment relates to the former comment of l. 514:**

**Conceptually, all ensemble members are equally accurate, given they are forced with the same set of external forcings. A thought experiment might be if one could estimate the outbreak of a volcano based on the single ensemble members and reconstruct volcanic activity. As one can see in the additional rainfall anomaly plots ensemble member 01, the event E01 shows a comparatively large response, whereas member 02 shows little or no response at all (even in the presence of the very strong E01 eruption). This might be also the case for a future volcanic eruption that despite a (clear) simulated ensemble mean response there is a chance on the regional scale for little or no response at all. Maybe the authors could just add some words to motivate a more nuanced view on the response of hydrological changes on even large volcanic eruptions, especially on the local-to-regional scale and the according disadvantages [of] only investigating ensemble mean statistics.**

R1.1. In our first round of revisions, we inserted several additional sentences based on these results, and relevant to the reviewer's comment, in the discussion sections. We have amended these sentences for clarity. See line 515, page 24 of the revised manuscript:

*"Any individual ensemble member might best represent the historical reality, but it is impossible to select the most accurate member absent supporting observational data from the period. Also, added noise due to natural variability can be greater at the regional scale, even to the extent of*

*altering the sign of observed changes among the individual ensemble runs. Thus, we mainly focused upon the mean from across the ensemble when examining the response to the eruptions for the various climate variables considered"*

We certainly agree with the substance of the reviewer's comment, and have additionally added the following text in our conclusion section (Page 38; Line 811).

*"However, we note that particularly on smaller spatial scales, as examined here, the variability in the modelled response as observed across our individual ensemble members may reduce the representiveness of the mean. The notable variability on display across our individual ensemble members, even to the quite substantial forcing represented by E1, also suggests that hydroclimatic responses on local to regional scales may depart from broader regional or hemispheric averages even after quite large volcanic forcings.*

**C1.2. The second comment is related to Line 743-785 In the Revised Manuscript:**

**The 1 sigma ctrl level is a very poor statistical threshold to argue for the statistical significance/confidence. Therefore the 2 sigma level, representing approximately the 95% confidence interval should be used. It is always advisable to rigorously test a hypothesis based on an a-priori set threshold of statistical significance instead of re-defining thresholds after the analysis is carried out to fit results to the according hypothesis and/or lines of argumentation. Therefore the presentation of results is now better, but implicitly also reveals the insight that the smaller northern hemispheric eruptions E02-E04 show a remarkably different response compared to the larger tropical eruption E01.**

R1.2. We have now modified the discussion starting at line 699 to remove the discussion pertaining to the 1σ level, keeping the discussion for the 95% (±2σ) only. We have also modified plot 12 to remove the line representing the 1σ, as shown below. We have additionally added the following sentence added in the conclusion at line number 699.

*"It is evident that the mean surface temperature response in the northern hemisphere is significant at the control period's $2\sigma_{ctrl}$ level (95% significance). However, while rainfall and river discharge responses are not significant at the $2\sigma_{ctrl}$ levels, several individual members do show significance at $2\sigma_{ctrl}$ as well. This signifies the important influence of the model's internal variability in representing the regional hydrological response to volcanic eruptions."*

[Figure]

Fig 12: Monthly time series of individual ensemble and mean of surface temperature response (˚C) averaged over northern hemisphere (NH) (top panel), rainfall change (mm/day) for the model's spatial box representing the Nile River watershed (Latitude: (5N, 18N), Longitude: (30E, 42E)) (middle panel) and Nile River discharge anomaly (%) at the delta region (grid box centered at 29.0N, 31.25E). For each panel, the darker solid (thick) line shows the multi-ensemble mean, individual member (thin line), and the color envelope shows the associated variability (±σ; Standard deviation). The annual cycle of climate variability of the control run is shown as $2\sigma_{ctrl}$ lines (red solid line) along the x-axis for all three variables. The vertical dotted green line shows when each eruption happened.

**Reviewer 3: -**

**Comment 1.0. The authors' changes have addressed most of my concerns in the first draft. The additions starting at line 275, in particular, help clarify the scope of the article.**

**Response 1.0.** We thank the reviewer for their assessment.

**C1.1. Nevertheless, the discussion of historical causation could still use improvements. In particular, the authors face a central problem that the correlation between the timing of eruptions and the timing of uprisings in Ptolemaic Egypt fits two scenarios equally well: (1) the occurrence (rather than non-occurrence) of eruption-induced droughts was necessary**

**(and/or sufficient) for the occurrence (rather than non-occurrence) of uprisings, or (2) the timing of eruption-induced droughts in some years (rather than droughts occurring in other years) was necessary (and/or sufficient) for the timing of uprisings in some years (rather than uprisings that would have taken place anyway in other years). This is not a fatal problem with the article. However, it needs to be openly admitted and addressed. The authors do not need to determine which scenario was the case, and they may clarify how the findings of this study add some credibility to scenario (1). However, they need to acknowledge that both scenarios remain possible and that some historical evidence points to (1) while other historical evidence and circumstances point to (2).**

**R1.1.** We appreciate the reviewer's efforts in outlining and explaining their position on the importance of (and approaches to) assessing historical causality. This is valuable (and again we are happy that *Climate of the Past* has an open peer review system whereby the reviewer's commentary can be accessed and cited). We are similarly happy to amend our Introduction to explicitly cite the model of necessary and sufficient condition (described simply) as useful for further interrogating and thinking about causality, and something that readers should be aware of.

See now Lines 143-145 in the Introduction (of the clean manuscript; line numbers will differ in the Track Changes version), which state:

*White and Pei (2020) argue that such questions represent a key challenge for climate historians and related scholars, recommending a framework wherein potential causes are assessed using a framework of necessary and sufficient conditions (put simply; see also Ludlow et al. (2023)).*

See also Lines 857-860, which close the paper:

*Relatedly, open questions remain as to where along the spectrum from proximate to ultimate causality (as per Gao et al., 2021) or necessary and sufficient conditions (as per White and Pei, 2020) hydroclimatic shocks lay in contributing to the revolts and other societal stresses that feature so prominently in Ptolemaic history.*

We argue that to go much further in discussing the intricacies of causality is, however, beyond the scope of the present study, which was never intended to engage in a full discussion of how historical causality should be assessed, much less to attempt to determine the precise nature of that causality (as, to be fair, the reviewer notes). To do so would unfairly divert the reader from the considerable effort and time that expended in the modeling exercise, presented as the core of the paper. It is true that this modeling has been conducted in pursuit of a greater insight into the likely hydroclimatic consequences of the eruptions between 168-158 BCE, which were chosen not only for their scientific interest in terms of apparent magnitude and close temporal spacing, but because they coincided with a period of pronounced political turmoil in Ptolemaic Egypt. But it is for other work (already in progress) to make use of the modeling insights presented in this paper to deliver a dedicated analysis of the role of these volcanically induced

hydroclimatic shocks in the incidence of any given social phenomenon (like revolt) of the period.

We do, however, explicitly note (as recommended by the reviewer) that the nature of any underlying causation is not settled: See Lines 139-141:

*Much work remains to further characterize this causality, how direct or indirect it may have been, and whether this changed meaningfully through time (and between revolts that varied in geography and scale) according to (or in interaction with) other coincident potential causes (from longer term developments promoting chronic vulnerabilities, to more acute political and socioeconomic stresses).*

**C1.2. I would particularly encourage revisions in two sections:**

**The authors' discussion of causation and correlation starting at line 183 is confusing. It is true that historians frequently resort to the platitude "correlation isn't causation." That platitude often applies to situations in conventional history, where a correlation between phenomena A and B might be explained away by some set of factors (C, D, E, etc.) that influence both A and B, rather than any connection between A and B themselves. However, in much climate history, where there are no hidden variables influencing, e.g., both social unrest and volcanic eruptions, that problem doesn't apply and we need to drop the platitude altogether.**

**R1.2.** We appreciate the reviewer's reasoning here, and have amended our relevant discussion to better frame and justify the inclusion of correlation vs. causality. We note that because the topic is familiar to many readers it thereby acts as an effective window into a brief discussion of causality, as well to introduce the work done to date by Ludlow and Manning (2016) and Ludlow et al. (2017) in establishing the statistical significance of the eruption-revolt association.

In Lines 124-135, we now state:

*It is a truism that correlation does not establish causation. Genuine causality is, however, implied where significance testing suggests an observed correlation is unlikely to have arisen randomly, though this does not determine the direction or character of causality (Izdebski et al., 2023). Statistical significance may, moreover, be sensitive to many factors. These include here (1) the choice of statistical test, (2) the choice of revolt and eruption dates (if uncertainties exist), (3) judgements as to what constitutes "revolt" (vs. phenomena like food riots motivated more by desperation than politics), and (4) judgements concerning which eruptions to include in testing (e.g., seeking those with a meaningful impact vs. non-climatically effective eruptions introducing "noise" into the analysis), assessed by estimated volumes and heights of atmospheric SO2 injections, eruption locations, and more. Notably, thus, testing by Ludlow and Manning (2016) was followed by Manning et al. (2017) who also observed a statistically significant coincidence between eruptions and Ptolemaic-era revolts using different methods and variant dates.*

**C1.3. In this study [it] would be more accurate to say that this correlation does imply causation—yet the nature of that causal linkage remains unclear. Absent further research and clarification, we cannot say, for example, whether these uprisings might have occurred without the eruptions but perhaps in a different manner or in a different year. Moreover, even if the eruptions and Nile failures were a necessary condition for the uprisings, we would need further information and arguments to determine whether the eruptions should be deemed "the cause of" the uprisings. For example, if the Ptolemaic regime were especially vulnerable to Nile failures at this time, while another regime would have endured similar natural shocks without popular unrest, then it may be more appropriate to label those societal vulnerabilities "the cause of" the uprisings instead.**

**The discussion of causal "pathways" does not necessarily address this problem and could be even more misleading. After all, almost every causal pathway could be broken down into additional causal mechanisms ad infinitum. The length or complexity of the "pathway" per se doesn't actually change how we determine causation. I can't run over someone in a car and then claim that my actions didn't cause their death because really there was this whole complex pathway between pressing my foot on the accelerator, the motor running, the car moving, the impact, injury, blood loss, etc. What matters, as discussed in the review of the first version, are determinations of the appropriate contrast set in cause and effect and the strength of causal necessity and sufficiency.**

R1.3. We appreciate the reviewer's continued argumentation and welcome the opportunity to engage with it in more depth here (given that we feel this would be too much of a diversion with the focus on modeling in the main text of the paper itself).

To begin, we agree that based upon the statistical testing of Manning et al. (2017) that the repeated coincidence in timing between historically dated revolts and ice-core-based eruption dates suggests causality. As we would describe it, the "direction" of the causal "pathway" cannot credibly flow in reverse here (i.e., revolts do not credibly cause eruptions). We have amended the text of our Introduction to emphasize this more clearly.

Lines 136-139 thus note (following on directly from the excerpt cited in our previous answer):

*Logic dictates that the direction of any genuine causality must flow here from eruption to revolt (Izdebski et al., 2023). Further confidence in its reality arises from the existence of plausible mechanisms connecting volcanic hydroclimatic variability with revolt (i.e., via reduction of the Nile summer flood and consequent societal impacts).*

Even in this case, it is not actually so straightforward to comprehensively reject concerns over correlation vs. causality. Doing so depends upon accepting the central finding of Manning et al. (2017), in which the coincidence in timing between revolts and eruptions is held to be real (i.e., causal) on the basis that it appears non-random (statistically significant), with the margin for error in terms of this statistical significance being deemed acceptably small (itself a value judgment that may vary by individual). Given that an individual's judgment as to whether the

observed coincidence in timing (correlation) is likely to arise from actual causation will depend in large measure upon the level of statistical significance observed, it is critical to note that this might change meaningfully depending upon (1) the type of statistical testing chosen (there are always alternative tests possible), (2) the set of revolt dates chosen in cases where dating uncertainty exists, (3) whether it is accepted that all events considered by Manning et al. (2017) should be taken to be the same type of phenomenon for the purpose of testing (after all, each revolt will have been to varying degrees unique in circumstances, severity, geography, duration, motives, outcomes, etc., so that other revolt groupings may be legitimately proposed), and (4) what eruption dates should be included as relevant for the purpose of this testing (e.g., which eruptions were likely, in principle, to have a meaningful hydroclimatic impact as based upon their apparent magnitudes, locations, etc., and not simply introduce "noise" to any analysis). We have noted these issues more briefly in the main article text, as per the excerpt provided earlier (Lines 124-135).

One critical aid to any assessment of correlation vs. causality (in light of the above) is the delineation of causal pathways that can credibly link proposed/potential causal forces or factors (like explosive volcanism) to a complex societal phenomenon (like revolt). As Manning et al. (2017) propose, one of several (non-exclusive and simplified) possible causal pathways linking revolt to explosive volcanism involves the demonstrable dependence (for its agricultural fortunes and even political stability) of Ptolemaic Egypt upon a sufficient Nile summer flood (noting that what was "sufficient" will certainly have varied in time – a point the reviewer touches upon in a related context - and which we emphasize in Lines 140-144), and the ability of large explosive eruptions to diminish the level of summer flooding. It is the intention of the present paper to provide insight into the extent to which historical eruptions like those observed between 168-158 BCE might have impacted the flood, and hence contribute to more detailed assessments of Egyptian environmental history (including causality). We emphasize this on Lines 186-189:

*Here, we intend to advance our understanding of the likely hydroclimatic impact of the 168-158 BCE eruption quartet as a foundation for ongoing efforts to more securely establish and qualify the causality underlying the observed association between eruptions, Ptolemaic-era revolts and other political and socioeconomic phenomena and developments.*

As is evident from our response, we hold that referencing casual or contributory pathways is a useful way to conceptualize causality (quite common in the contemporary climate-conflict literature and thereby a useful reference point for readers coming from this field) and for thinking about the dependencies between different historical phenomena, including in our own case. Whilst we accept the reviewer's point that one could add additional contributory mechanisms "ad infinitum", we would argue for the utility in an exercise and framework that attempts to identify and order multiple potential causal / contributory forces and factors in trying to explain human history. Moreover, we do not see this as mutually exclusive to the framing of necessary and sufficient conditions favored by the reviewer. Thus, on Lines 151-156, we now state:

*An alternative framing in many climate-conflict studies (not incompatible with that proposed by White and Pei (2020) or employed by Gao et al. (2021)) is to delineate multiple identifiable "pathways" that may enable or lead (through material (economic), political, cultural or psychological channels) to links between hydroclimatic variability and various forms of conflict (see Hsiang and Burke (2014 ) and Ide (2017) for reviews).*

For the record, we have discussed the importance of many such considerations as they pertain to causality (in a specific case study on Ptolemaic Egypt) in a paper already in-press: Izdebski, A., Bloomfield, K., Eastwood, W. J., Fernandes, R., Fleitmann, D., Guzowski, P., Haldon, J., Ludlow, F., Luterbacher, J., Manning, J. G., Masi, A., Mordechai, L., Newfield, T., Stine, A. R., Senkul, C. and Xoplaki, E. (In Press, 2023), "The Emergence of Interdisciplinary Environmental History: Bridging the Gap between the Humanistic and Scientific Approaches to the Late Holocene," *Annales*, 77 (2), 1-48. We have also applied different tests using different lists of eruptions and revolts and found a continued statistical significance in the observed coincidence between revolt dates and eruption dates. See Ludlow, F. and Manning, J. G. (2016) "Revolts under the Ptolemies: A Paleoclimatic Perspective", In: Collins, J. J. and Manning, J. G. (eds.), *Revolt and Resistance in the Ancient Classical World and the Near East: The Crucible of Empire. Culture and History of the Ancient Near East Series*. Leiden: Brill, 154-171.

**C1.4. I would also discourage the use [of] "proximate" vs. "ultimate" causation in lines 1303-1305 for the following reasons. First, the terms themselves are confusing and ambiguous. Without further explanation, many readers would assume that an "ultimate" cause should be somehow more fundamental than a "proximate" cause—exactly the opposite of how they are used in Gao et al. Second, the long time-series and abundant data for disasters and political events in imperial China enable inferences and analysis that just aren't possible (yet) for a case like Ptolemaic Egypt. Third, and most important, the spectrum between "ultimate" and proximate" fails to capture the central problems regarding causation in the case of Nile failures and uprisings in Ptolemaic Egypt. As explained in the review of the first draft, these problems are essentially two-fold. On the one hand, we cannot say (yet) whether these Nile failures were necessary or sufficient for the uprisings to occur at all, or only whether they were necessary or sufficient for the timing or perhaps character of social and political turmoil that was going to occur sooner or later anyway. On the other hand, even if eruption-induced Nile failures were necessary or sufficient for these events, we cannot say (yet) which condition was more exceptional and relevant and therefore appropriately labeled "the cause of" the uprisings: the degree of Nile failure or the particular vulnerabilities of Ptolemaic regime.**

**I would strongly encourage the authors to spell out these issues of historical causal explanation plainly and clearly.**

**R1.4.** As noted earlier, following the reviewer's guidance, we have added text highlighting the framework of necessary and sufficient conditions to the Introduction, where it is presented as one useful framework for further interrogating the issue of causality. But given that the matter of how to assess causality is hardly settled in or between various disciplines, we argue (similar

to the "pathway" framing) that we should keep the reference to "ultimate" versus "proximate" causality. Beyond Gao et al. (2021), it is finding increased use as a framework to further interrogate or characterize historical causality (see, e.g., Brian Villmoare, *The Evolution of Everything: The Patterns and Causes of Big History* (Cambridge University Press, 2022), in which ultimate and proximate causality are employed as central concepts - we have now cited this in the paper). It may well be that one approach or framing can ultimately be demonstrably shown as better than another, but the present paper is not the place to discuss this in depth. Rather, in our revisions, we now offer a range of such framings, for the reader to further examine.

We do take the reviewer's point that some readers may assume that "ultimate" should be taken as the more important form of causality and have clarified that in the existing text. See now Lines 145-150:

*Gao et al. (2021) employ a framework wherein the role of volcanically induced hydroclimatic "shocks" in the collapse of Chinese dynasties is characterized along a spectrum from "ultimate" to "proximate" causation (see also Villmoare (2022) for this framework). Here, smaller hydroclimatic shocks could act as the ultimate cause of collapse when enabled by high pre-existing stress, while larger shocks could act with greater independence as proximate causes without substantial pre-existing stress.*

**1.4. Finally, I would encourage the authors to tighten the language to improve readability, particularly in sections that have been added since the first draft (e.g., lines 180-183). The article is still unusually long. While the complexity of the topic does merit extra space, I believe it could be at least a page shorter simply by improving the style and removing ambiguities and redundancies.**

We agree that greater concision can be achieved and have made many further excisions and tweaks to streamline the text for clarity and size (please see the Track Changes version of the manuscript). This includes the deletion of several paragraphs outright (e.g., see the Discussion and Conclusions section).

Regarding length, we note that addressing the issue of causality has itself contributed to a longer article, but feel the additions and overall length are justified here because (1) the interdisciplinary subject matter requires more context to guide its different potential audiences, (2) there are no set word limits (that we are aware of) in this journal, and (3) this paper is certainly not the longest of those already published in this special issue of *Climate of the Past*.

---

## Author Response (AR3)

**Response to Final Editorial comments:**

Investigating hydroclimatic impacts of the 168-158 BCE volcanic quartet and their relevance to the Nile River basin and Egyptian history

Singh et al. 2022.

We are grateful to all reviewers and editor for their valuable time, comments and suggestions for improving the manuscript. We believe these have helped us further improve the manuscript. The responses to the editorial corrections are in bold and text added to revised manuscript is in italics.

**Comments: -**

L19: "strato-volcanic": here I think you mean to refer to eruptions that inject sulfur to the stratosphere, but since stratovolcano is a kind of volcano, this can be confusing. I would suggest to remove the word.

**Response: - we removed the word "strato" and now it read as "volcanic eruption".**

L48: Here you refer to the global total aerosol optical depth increasing from about 0.6 to 0.75 after the Pinatubo eruption, numbers which include tropospheric as well as stratospheric aerosol. It is probable that many readers will not recognize that the tropospheric component is included here, and since later analysis focuses specifically on stratospheric aerosol, I would suggest considering phrasing this sentence in terms of stratospheric aerosol alone.

**Response: - Rephrased the sentence as suggested. (L48)**

L116: "statistically significant"

**Response: - corrected as suggested at line number 120.**

L321: change to "ka", i.e., lowercase "k" for "kilo"

**Response: - corrected as suggested.**

L322, and throughout: fig-> Fig.

**Response: - corrected throughout the manuscript.**

L331: simulated a global mean surface...
**Response: - corrected. L336**

L387: Since you've given the LW anomalies as positive values above, and report the net anomalies as negative numbers below, it would make sense to report the SW anomalies as negative values here.

**Response: - corrected.**

L441: Most of the observations reported by Russel et al. (1996) are from specific locations, while what you show in Fig. 6 is a global mean. This may limit the degree to which you can compare the two, which may be worth a comment.

**Response: - We have added information in the sentence which highlights this important point while putting in a comparison for aerosol size. Given below. Line 453**

**"In comparison, $R_{eff}$ after Pinatubo (1991) increased to 0.6 μm as observed at a number of specific locations reported by Russell et al. (1996), with that size sustained for approximately 2 years. By contrast, the model simulated global sulfate aerosol sizes for the three subsequent extratropical eruptions (E2 to E4) grew up to 0.3 μm."**

L451: Not sure what you mean by "phase of" the Brewer Dobson Circulation here. To be picky, the BDC is a wave-driven circulation, so the aerosol heating changes in dynamics don't affect the BDC, they are superimposed upon it. So, I would suggest to edit down this sentence removing reference to the BDC.

**Response: - We modified the particular sentence, and "phase of Brewer Dobson Circulation" is replaced by the more general "atmospheric circulation". Line: 466.**

L470: I would ask the authors to be careful here: was there really no increase in AOD in the SH, or was it only small compared to that of the NH? Some modeling studies have found increases in AOD in the opposite hemisphere around 10% of that of the hemisphere of eruption, which may be significant if the eruption is large enough.

**Response: - Thank you for this comment. We have modified the sentence and mentioned the residual impacts noticed after the eruption E1 (in Fig 7A). Line: 478.**

**"However, the other three eruptions (E2, E3 and E4) in the high latitude Northern Hemisphere yielded an increased AOD primarily confined to that hemisphere, with the cross-equatorial AOD response maintaining the residual impact after tropical eruption E1."**

L491: I've only ever seen the term "Hovmoeller diagram" used to describe plots which plot an atmospheric variable as a function of longitude and time. Wikipedia suggests a more general definition, but says they are used to highlight the behaviour of waves. That does seem to be the goal of the first hovmoeller diagrams. See https://www.e-education.psu.edu/meteo820/node/546. I would suggest removing the reference here and throughout to avoid confusion.

**Response: - Thank for pointing this out. We agree that Hovmöller diagram is basically designed to show the wave like pattern though longitude and time. We have decided to rename it to "Hovemöller plot" instead of diagram to avoid confusion, as in recent times this plot has been used to show the behavior of a variable by latitude and time. We have also characterised it as a "Hovmöller type plot (7A & B)…" in the beginning of section 3.4. Another reason to keep the reference to Hovmöller is that we have used this terminology across the various iterations of the paper throughout the revision and review process, so we have decided to keep it with a slightly amended name for the shake of consistency.**

L500: Well, the anomalies in Fig 7C seem largely constrained to be within +/- 60degrees, which is not exactly "equatorial".

**Response: - We have modified the sentence to better characterize this.**

**"*Fig. 7C shows that the latitudinal anomaly of the lower stratosphere warming was centered along the equator and largely constrained 60N-60S.*"**

L503: The impact of volcanic forcing on tropospheric dynamics through the stratospheric temperature gradient has become more controversial in recent years, see e.g., https://acp.copernicus.org/articles/20/13687/2020/acp-20-13687-2020.html. I don't think a citation or discussion is needed here, just perhaps a softening of the language to better reflect the lack of consensus on this issue.

**Response: - We modified the sentence and included this as reference for scant evidence.**

**"*Lower stratosphere warming is also thought to affect the northern hemisphere atmospheric circulations, though efforts to confirm the mechanisms and consistency of this response are ongoing (e.g., Graf et al., 1993, 2007; Shindell et al., 2004; Polvani and Camargo 2020).*"**

**"*Polvani, L. M. and Camargo, S. J.: Scant evidence for a volcanically forced winter warming over Eurasia following the Krakatau eruption of August 1883, Atmos. Chem. Phys., 20, 13687–13700, https://doi.org/10.5194/acp-20-13687-2020, 2020.*"**

L507: Courser than what?

**Response: - we have changed our phrasing to now read: "*we used a relatively coarse resolution model ….*"**

L508: what kind of changes is it successful in simulating? Response to volcanic forcing? Or GHGs? Or something else?

**Response: - We have modified this sentence for the introduction of GISS modelE and added relevant references.**

*"We used a relatively coarse resolution earth system model having a simplified parameterization that is skilled in simulating the large-scale patterns of climate response to natural and anthropogenic forcings (Kelley et al., 2020; Miller et al., 2021; Nazarenko et al., 2022)"*

*Refs:*

*Larissa S. Nazarenko, Nick Tausnev, Gary L. Russell, David Rind, Ron L. Miller, Gavin A. Schmidt, Susanne E. Bauer, Maxwell Kelley, Reto Ruedy, Andrew S. Ackerman, Igor Aleinov, Michael Bauer, Rainer Bleck, Vittorio Canuto, Grégory Cesana, Ye Cheng, Thomas L. Clune, Ben I. Cook, Carlos A. Cruz, Anthony D. Del Genio, Gregory S. Elsaesser, Greg Faluvegi, Nancy Y. Kiang, Daehyun Kim, Andrew A. Lacis, Anthony Leboissetier, Allegra N. LeGrande, Ken K. Lo, John Marshall, Elaine E. Matthews, Sonali McDermid, Keren Mezuman, Lee T. Murray, Valdar Oinas, Clara Orbe, Carlos Pérez García-Pando, Jan P. Perlwitz, Michael J. Puma, Anastasia Romanou, Drew T. Shindell, Shan Sun, Kostas Tsigaridis, George Tselioudis, Ensheng Weng, Jingbo Wu, Mao-Sung Yao, Future Climate Change Under SSP Emission Scenarios With GISS-E2.1, Journal of Advances in Modeling Earth Systems, 10.1029/2021MS002871, 14, 7, (2022).*
*Miller, R. L., Schmidt, G. A., Nazarenko, L. S., Bauer, S. E., Kelley, M., Ruedy, R., Russell, G. L., Ackerman, A. S., Aleinov, I., Bauer, M., Bleck, R., Canuto, V., Cesana, G., Cheng, Y., Clune, T. L., Cook, B. I., Cruz, C. A., Del Genio, A. D., Elsaesser, G. S., Faluvegi, G., Kiang, N. Y., Kim, D., Lacis, A. A., Leboissetier, A., LeGrande, A. N., Lo, K. K., Marshall, J., Matthews, E. E., McDermid, S., Mezuman, K., Murray, L. T., Oinas, V., Orbe, C., Pérez García-Pando, C., Perlwitz, J. P., Puma, M. J., Rind, D., Romanou, A., Shindell, D. T., Sun, S., Tausnev, N., Tsigaridis, K., Tselioudis, G., Weng, E., Wu, J., and Yao, M. S.: CMIP6 Historical Simulations (1850–2014) With GISS-E2.1, J. Adv. Model. Earth Sy., 13, e2019MS002034, https://doi.org/10.1029/2019MS002034, 2021.*

L520: See comment on Hovmoeller above.

**Response: - modified as per previous comment**

L524: Hovmoeller

**Response: - modified as per previous comment**

L529: This looks less like a trend than a persistent negative anomaly

**Response: - we have modified this sentence as given below (Line 540).**

*"A robust negative anomaly on the order of 0.3-0.4 mm/day in the northern hemisphere rain belt (ITCZ) region appeared during the next year following E1, with a persistent negative anomaly in the subsequent years (Fig. 8)"*

L532: at the equator

**Response: - Corrected. Line 566.**

L641: The usual multiplier for a 95% CI is 1.96*sigma, not 1.95.

**Response: - yes, it is approximately 1.96. We used 1.95 as suggested by one of the reviewers during the review process.**

L642: Are the values in the table the ensemble standard deviation and ensemble mean? Table 2: the "Change/Std" labels in the column heading could be interpreted as meaning the values are divided by (normalized by) the standard deviations, which I think is not the case. Please modify the column headings to avoid potential misunderstanding.

**Response: - We have modified it according to the values tabulated here as Change±Std.**

Fig 11: use lowercase k in "km" in label to colorbar

**Response: - Corrected**

L692: use degree symbols in latitude rages

**Response: - Corrected**

L732: ensemble members

**Response: - Corrected**

L759: the eruptions aren't identified, I would suggest "detected"

**Response: - Corrected**

L838: ensemble members

**Response: - Corrected**

L854: "driving"?

**Response: We have changed this to "bringing" for greater clarity.**

L871: "on request to the corresponding author"?

**Response: - corrected**

Fig S1: what quantity is shown in the plots, is it changes in fractional area covered by each plant type?

**Response: - We have modified the figure caption to include the details of quantity.**

Fig S3: Please indicate in the caption here that these are arbitrarily chosen locations.

**Response: - Modified to include this information.**

Fig S6: This plot has a distracting patterning of the stipling which should be redrawn.

**Response: - We have modified the plot by redrawing the stippling in the manner used for the rainfall plot.**